# Brain aging comprises many modes of structural and functional change with distinct genetic and biophysical associations

Stephen M Smith[1]*, Lloyd T Elliott[2], Fidel Alfaro-Almagro[1], Paul McCarthy[1], Thomas E Nichols[1,3], Gwenaëlle Douaud[1], Karla L Miller[1]

[1]Wellcome Centre for Integrative Neuroimaging (WIN FMRIB), University of Oxford, Oxford, United Kingdom; [2]Department of Statistics and Actuarial Science, Simon Fraser University, Vancouver, Canada; [3]Big Data Institute, University of Oxford, Oxford, United Kingdom

**Abstract** Brain imaging can be used to study how individuals' brains are aging, compared against population norms. This can inform on aspects of brain health; for example, smoking and blood pressure can be seen to accelerate brain aging. Typically, a single 'brain age' is estimated per subject, whereas here we identified 62 modes of subject variability, from 21,407 subjects' multimodal brain imaging data in UK Biobank. The modes represent different aspects of brain aging, showing distinct patterns of functional and structural brain change, and distinct patterns of association with genetics, lifestyle, cognition, physical measures and disease. While conventional brain-age modelling found no genetic associations, 34 modes had genetic associations. We suggest that it is important not to treat brain aging as a single homogeneous process, and that modelling of distinct patterns of structural and functional change will reveal more biologically meaningful markers of brain aging in health and disease.

*For correspondence: steve@fmrib.ox.ac.uk

## Introduction

Brain imaging can be used to predict 'brain age' - the apparent age of individuals' brains - by comparing their imaging data against a normative population dataset. The difference between brain age and actual chronological age (the 'delta', or 'brain age gap') is often then computed, providing a measure of whether a subject's brain appears to have aged more (or less) than the average age-matched population data. For example, looking at structural magnetic resonance imaging (MRI) data, a high degree of atrophy would cause a subject's brain to appear older than a normal age-matched brain. Estimation of brain age and the delta is of value in studying both normal aging and disease, with some diseases, such as Alzheimer's disease, showing similar patterns of change to that of accelerated healthy aging (*Franke et al., 2010*; *Cole and Franke, 2017a*; *Cole et al., 2017b*).

The typical approach uses one or more imaging modalities, most commonly using just a single structural image from each subject. The data is then preprocessed, and features identified, for use in the brain age prediction. For example, the structural images may be warped into a standard space, and grey matter segmentation carried out; the voxelwise segmentation values themselves can then be the features. Alternatively, a smaller number of more highly condensed features may be derived, such as volumes of grey and white matter within multiple brain regions. The resulting dataset, of multiple subjects' feature sets, along with their true ages, is then passed into a supervised-learning algorithm (e.g. regression, support vector machine or deep learning). The algorithm then learns to predict the subjects' ages from their brain imaging features. Finally, the true age is typically

subtracted from the estimated brain age, to create a delta, potentially with corrections for biases such as systematic mis-estimation of brain age (*Le et al., 2018*; *Smith et al., 2019*).

The imaging feature set can be derived from more than one imaging modality, in which case it can contain information not just about the structural geometric layout of the brain, but also, for example, structural connectivity, white matter microstructure, functional connectivity, iron deposition, and cognitive task activation (*Brown et al., 2012*; *Liem et al., 2017*; *Richard et al., 2018*; *Vinke et al., 2018*; *Groves et al., 2012*). Such 'multimodal' data allows for brain age modelling to take advantage of a richer range of structural and functional measures of change in the brain, but it is still the case that most brain-age modelling only estimates a single overall brain age per individual.

Hence, while the explicit goal of much brain-age research is to obtain a single estimate of brain age (and brain-age delta) per subject, one could nevertheless expect that multiple distinct biological processes contribute to the changes seen in the brain with aging. For example, amounts of physical exercise, intake of alcohol and smoking, dietary patterns, and health factors such as hypertension and obesity, will all likely contribute to the 'aging' of the brain, and in potentially different ways. These different factors will likely affect different aspects of the brain's structure and function, as viewed through multiple imaging modalities. Further, different factors affecting brain aging could well have different age dependence - population-averaged aging curves for the different factors could be quite distinct (e.g. with respect to strength and linearity of the age dependence) (*Kessler et al., 2016*; *Brown et al., 2012*; *Richard et al., 2018*; *Vinke et al., 2018*; *Douaud et al., 2014*; *Groves et al., 2012*). Different biological factors of brain aging might well also be expected to show distinct genetic influence. The combination of all factors into a single estimate of brain age can be a useful, compact, single summary metric, and is by definition the route by which the most accurate single estimate of a subject's age can be predicted from the imaging data available. However, this may come at the cost of losing important information regarding the distinctions between multiple biological factors occurring, making it harder to understand the (potentially multiple) causes of brain aging.

Here, we used six brain imaging modalities from UK Biobank (*Miller et al., 2016*) to identify 62 distinct modes of population variation, almost all of which showed significant age effects. In this work, we focus on investigating the distinct modes as potentially representing distinct biological factors relating to aging. We aimed to learn about a larger number of distinct modes, and in greater biological depth, than had been previously possible, in part because of the richness of the imaging and non-imaging data available in UK Biobank, and of course due in part to the very large subject numbers. There is nevertheless a link between this approach and the previous literature; one can combine the population modes to produce a single brain-age estimation, which gives similar age prediction accuracy to that derived using standard approaches.

We used the multimodal brain imaging data from 21,407 participants, over the age of 45y, in UK Biobank. Imaging is taking place at four sites, with identical imaging hardware, scanner software and protocols (although the subjects used here were from the first two sites). The dataset also includes genetics, lifestyle, cognitive and physical measures, and health outcome information from the healthcare system in the UK. For this work we used 3913 IDPs (imaging-derived phenotypes, generated by our team on behalf of UK Biobank, and made available to all researchers by UK Biobank). The IDPs are summary measures, each describing a different aspect of brain structure or function. IDPs include functional and structural connectivity between specific pairs of regions, localised tissue microstructure and biological makeup, and the geometry of cortical and subcortical structures.

For our work here, rather than simply feeding all IDPs into one brain age model (e.g. regularised multiple regression), we first identify multiple modes that represent different combinations of IDPs that co-vary across subjects. We then use each of these modes separately in simple but standard brain-age modelling. The result is a large number of distinct brain age predictions for each subject, with the goal of each representing a different biological process. We now summarise our approach briefly.

After removal of imaging confound effects (see Materials and methods for details), we used independent component analysis (ICA *Hyvärinen, 1999*) to decompose the entire IDP data matrix of $N_{subjects} \times N_{IDPs}$ into 62 distinct modes of population variation (*Kessler et al., 2016*; *Elliott, 2018*). Each mode is described by two vectors. The first is a set of IDP weights, describing which specific aspects of brain structure and function (i.e. which IDPs) are involved in that mode (e.g.

a given mode might reflect volume of grey matter across various regions involved in language processing). The second is a set of subject weights (one value per subject), describing where in the population distribution a subject lies, in terms of strongly expressing a given mode of variation (e.g. a given subject might have considerably less grey matter in language regions than the population average). These subject-weight vectors (one vector per mode) can be used to help understand the biological meaning of, and causal factors behind, the modes of population variation, by computing associations with non-imaging factors and genetics (a genetic or early-life factor that correlates, across subjects, with our hypothetical mode might suggest biological causes of changes in grey matter volume in the language network). Here we use the subject-weight vectors to study brain aging; virtually all modes show a significant aging effect (*Figure 1*), and in this work, we study the different aspects of brain aging represented by the 62 modes (as well as 6 clusters of these modes).

Having identified these modes, our modelling of brain age for individual modes follows the same form as commonly used for brain age modelling. We predict subjects' actual age using a given mode's subject-weights-vector, and then subtract the age from the predicted age to obtain the mode-specific brain-age delta. We then use this in our association tests against non-imaging variables and genetics. Hence, instead of using all available data from the brain imaging to obtain a single ('all-in-one') estimate of brain age (and associated delta), we investigate brain aging for each mode separately, to capitalise on the distinct richness of information obtained within separate modes. An indication of the usefulness of doing this can be seen from the fact that many of the modes' delta estimates have significant genetic association (i.e. genetic factors that are significantly driving that aspect of brain aging). By comparison, the all-in-one estimate of brain-age based on a linear combination of modes combines across so many different biological factors that there is no significant, replicated genetic association for the all-in-one delta, despite the overall prediction providing a more accurate estimate of subjects' ages than any one individual mode.

All data are available upon application to UK Biobank. In addition to the main and supplementary figures in this paper, further material is available from the https://www.fmrib.ox.ac.uk/ukbiobank/BrainAgingModes website (see Data availability). This includes: all code written for the work described here; detailed figures, with individual modes' separate genome-wide association study (GWAS) Manhattan plots and resting-state functional MRI (rfMRI) summary brain images; all GWAS summary statistic files; spreadsheets listing all modes' IDP weights, associations with non-imaging-non-genetic variables and peak GWAS associations; and additional genetic analyses including functional annotation, gene expression, associated traits from previous GWAS studies, and genetic heritability/co-heritability results.

## Results

### Multiple modes and mode-clusters of brain aging

After discarding outlier data and subjects with high levels of missing/outlier imaging data, we retained data from 18,707 subjects (see Materials and methods). Split-half reproducibility testing ($P<10^{-6}$) resulted in estimation of 62 robustly-present ICA modes of population variation. For convenience (and without loss of generality), the modes were inverted where necessary in order for their correlation with age to be positive, and were re-ordered according to decreasing variance explained by a cubic model of age, as reflected in the inset plot of age-mode correlations in *Figure 1*. The figure shows the cubic fit of each mode as a function of age (later plots show these fits in more detail and quantitation). The majority of the modes show similar behaviour for females and males, but a few notable exceptions can be seen in supplementary figures (*Figure 1—figure supplements 3–9*), as discussed in more detail below.

Using all 62 modes together in an 'all-in-one' prediction of overall brain aging, mean absolute delta (the 'error' between age and predicted age) was 2.9y. As described in Materials and methods, the all-in-one model is a weighted sum of the 62 modes, where the weight for a given mode is a scalar value that is entirely driven by the unique variance of that mode ($\beta_i$ for mode $i$). This unique variance is also referred to as the 'partialled' mode, which is calculated by taking a mode's subject weight vector and regressing out the subject vectors of all other modes. Because these partialled modes isolate the unique subject variance described by a given mode, it is of interest to examine their associations with non-imaging variables, and similarly the associations of partialled deltas.

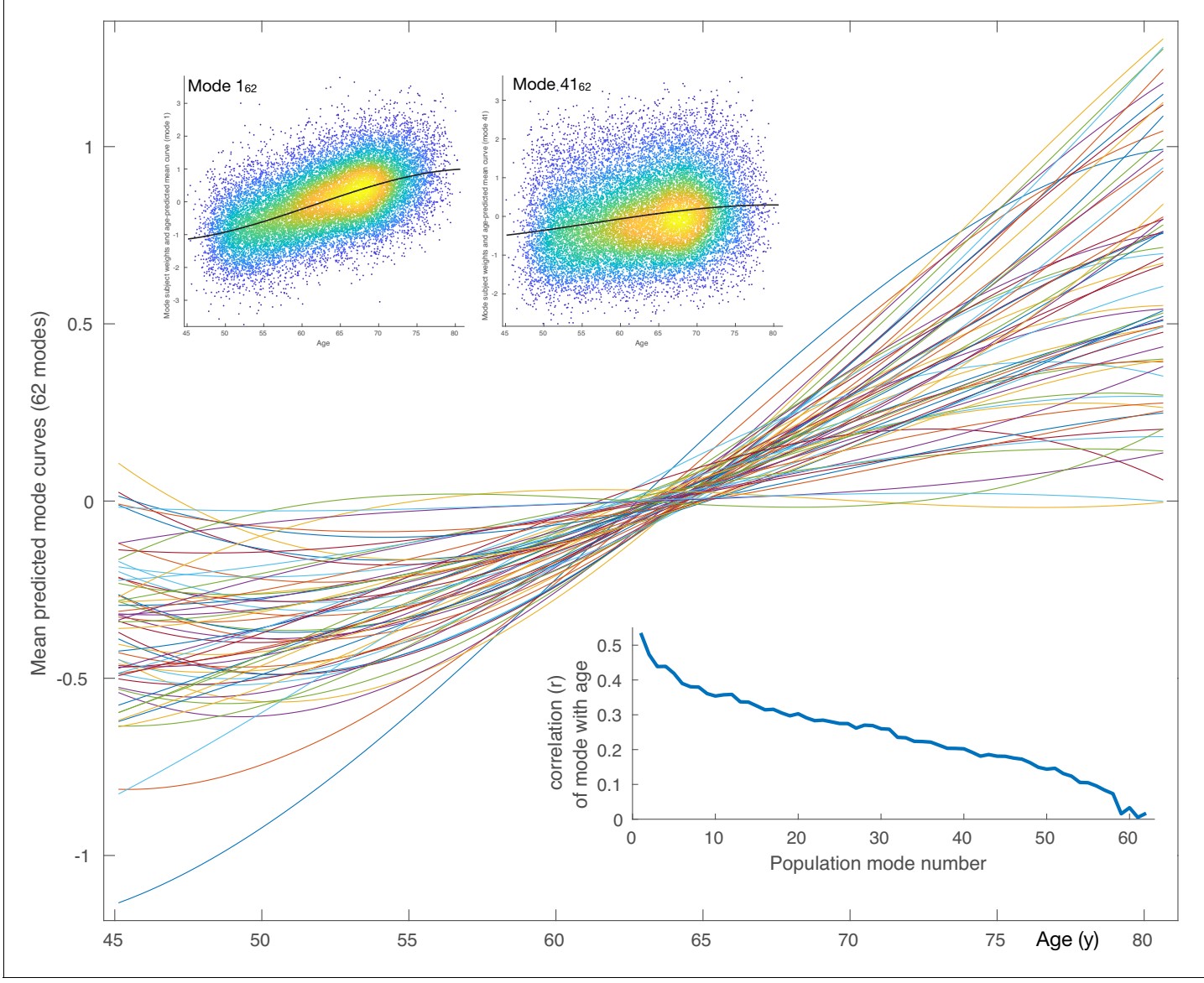

**Figure 1.** Mean aging curves for the 62 brain-aging modes. The main plot shows the mean aging curves based on a cubic age model - that is, fitting the subject-weight-vectors from each mode as a function of age, age-squared and age-cubed. Therefore, the x axis is age in years, and the y axis is the unitless values in the original modes' subject-weight-vectors $X_i$. The scatter plots show two example modes, with their respective mean aging curves shown along with the full data (the modes' subject weights, with a single point for each subject). The inset blue plot shows the strength of age prediction for all modes, quantified simply as correlation of actual age with mode subject-weights.

The online version of this article includes the following figure supplement(s) for figure 1:

**Figure supplement 1.** Hierarchical clustering of the 62 brain-aging modes, and their mapping onto six lower-dimensional mode-clusters.

**Figure supplement 2.** Model standard deviations, age correlations and age regressions for all modes and mode-clusters.

**Figure supplement 3.** Sex-separated mean age curves for modes 1–12.

**Figure supplement 4.** Sex-separated mean age curves for modes 13–24.

**Figure supplement 5.** Sex-separated mean age curves for modes 25–36.

**Figure supplement 6.** Sex-separated mean age curves for modes 37–48.

**Figure supplement 7.** Sex-separated mean age curves for modes 49–60.

**Figure supplement 8.** Sex-separated mean age curves for modes 61–62.

**Figure supplement 9.** Sex-separated mean age curves for mode-clusters 1–6.

**Figure supplement 10.** Non-additive modelling of brain-aging.

Hence, as seen in *Figure 1—figure supplement 2D*, the contribution to age modelling varies highly from mode to mode, driven by the unique variance in each. Several modes have negative $\beta$ weights, meaning that their unique variance is negatively associated with age, even though their original correlation with age was assigned to be positive. Of the 62 modes, 59 correlate significantly with age (at the $P<0.05/62$ two-tailed Bonferroni-corrected level), and 29 have a $\beta$ that is significant (i.e. their unique variance has significant age dependence).

In order to help generate more parsimonious descriptions of the 62 modes of brain aging, we investigated whether clustering modes together into a smaller number of mode-clusters could provide a meaningful simplification. Quantitative optimisation of the clustering dimensionality resulted in a meaningful reduction to six mode-clusters (see Materials and methods and *Figure 1—figure supplement 1*). As with the modes, mode-clusters were defined to correlate positively with age, and sorted in order of decreasing age dependence. As one might expect, there is less redundancy across these 6 mode-clusters (than across the 62 modes), for example, as shown by the fact that the genetic profiles for the partialled 6 mode-cluster deltas are similar to the non-partialled equivalents (*Figure 3—figure supplement 1*). For clarity, we refer to mode numbers using subscript '62', and to mode-clusters with subscript '6'.

## Mapping of brain-aging modes onto brain structure and function

*Figure 2* summarises the mapping of modes onto IDPs (different aspects of the brain's structure and function). Each row represents a mode/mode-cluster, and the 3,913 IDPs are arranged into distinct groupings as denoted within the figure. Within each grouping, each individual column represents a soft-clustering of highly correlated IDPs that have similar behaviour to each other (a complete list of the strongest associations between all modes and all IDPs is linked to in Data availability). In most cases, individual modes are largely driven by IDPs from a single imaging modality, with a few exceptions such as mode $52_{62}$. Naturally, the mode-clusters mix more across modalities. More specific discussion of individual mode and mode-cluster results are given below, in the context of the full set of imaging, non-imaging and genetic associations.

## Genome-wide associations studies of all brain-aging modes

We carried out a separate GWAS for the brain-aging delta from each of the 62 modes, and from the 6 mode-clusters. GWAS used 9,812,242 SNPs (single-nucleotide polymorphisms) that passed all quality control tests (see Materials and methods). We also carried out GWAS for two 'all-in-one' multiple-regression-based estimates of brain-aging delta, one using all 3913 IDPs in a single prediction of brain aging (with 55-dimensional principal component analysis, PCA, pre-reduction *Smith et al., 2019*), and the other using the 62 modes together (see Materials and methods). The GWAS paradigm we used was similar to that in *Elliott (2018)*, and associations were tested between these modes and 9,812,242 genetic variants. Results are summarised in *Table 1* and *Figure 3*. More detailed plots, including separate plots for every mode's GWAS, are provided in *Figure 3—figure supplement 1* and Data availability.

From the 62 GWAS of modes of brain aging, we found 156 peak associations passing the standard single-GWAS threshold of $-\mathrm{Log}_{10}P=7.5$, from the discovery sample of 10,612 subjects (*Figure 3A*). Here, 'peak associations' means that, in a region of high linkage disequilibrium (LD), we only report the SNP with the highest association with the phenotype, as the associations in the local region are most likely all due to a single genetic effect (see Materials and methods). 68 of these associations passed the more stringent threshold of 9.33, which increases the standard threshold by a Bonferroni factor of 62+6 to account for the multiple phenotypes' testing. From the smaller replication dataset of 5340 subjects, 64 of the 68 peak SNP associations replicated at the $P<0.05$ level. Of the 62 modes, 34 have at least one significant association at the higher threshold, and all these 34 modes have at least one association in the replication sample.

From the 6 mode-clusters, 14 regions of the genome have significant associations at the higher threshold, 12 of which replicate. Three of the these 6 mode-clusters have at least one significant association, including in replication.

The numbers of associations are lower for the partialled deltas (that reflect unique brain-aging profiles), with the numbers of significant associations approximately halving for the 62 modes, but being reduced only a small amount for the 6 mode-clusters (*Table 1*).

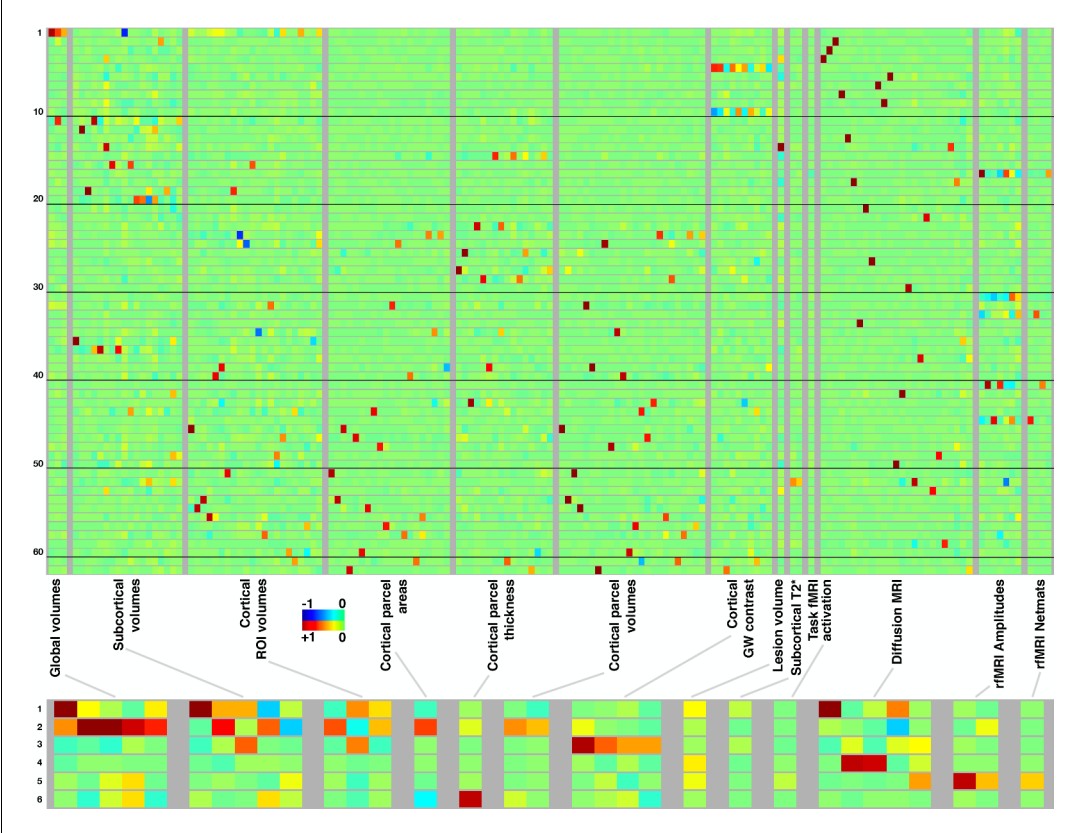

**Figure 2.** Mapping of 62 brain-aging modes and 6 mode-clusters onto different classes of strucural and functional imaging-derived phenotypes (IDPs). Above: Each row shows the mapping of one brain-aging mode onto the imaging data, with black lines delineating groups of 10 modes for ease of reference. The full plots spanning all 3913 IDPs are shown in *Figure 2—figure supplement 1*; here, each class of IDPs is reduced using PCA and then ICA to the most representative pseudo-IDPs (see Materials and methods), meaning that each column in the plot relates to a fixed and distinct combination of original IDPs. IDP classes have fewer/greater distinct values here dependent on the number of IDPs in a class, and how highly they correlate with each other. Colour-coded values shown are unitless and mapped into the range −1:1. Below: The equivalent (separately computed) summary figure mapping the 6 mode-clusters onto IDPs.

The online version of this article includes the following figure supplement(s) for figure 2:

**Figure supplement 1.** Mapping of brain-aging modes and mode-clusters onto individual IDPs.

**Figure supplement 2.** Histogram of proportions of subjects of (non-missing) data for each nIDP (non-imaging-derived phenotypes).

We also evaluated genetic associations for two 'all-in-one' estimations of a single best estimate of brain-age (and associated delta); we used all IDPs in one case, and all modes in the other. This was done with the methods described in *Smith et al. (2019)*. These two all-in-one brain-age delta estimations showed no genetic associations that were significant and replicated, consistent with previous GWAS of all-in-one brain-aging modelling (*Ning et al., 2018*). This suggests that biological specificity driving the mode/mode-cluster results has been lost (diluted) when generating a single brain-age delta.

Finally, estimates of genetic (SNP) heritability showed that 57 of the 62 modes were significantly heritable, as were all 6 mode-clusters (see Materials and methods and online supplemental results). Estimates of co-heritability with Alzheimer's and Parkinson's disease showed a small number of nominally significant results, but none of these survive multiple comparison correction across modes; this suggests that none of these modes of aging map strongly onto these diseases genetically.

## Associations of modes with non-imaging variables

We also computed associations between all modes' deltas and 8787 nIDPs (non-imaging-derived phenotypes), spanning 16 groups of variable types. These groups include early life factors (e.g. maternal smoking, birth weight), lifestyle factors (e.g. exercise, food, alcohol and tobacco variables),

**Table 1.** Summary results of all GWAS of brain-age delta estimates: numbers of supra-threshold SNP clusters from GWAS of all modes (discovery N = 10,612; validation N = 5,340).

Phenotypes fed into GWAS are grouped and reported on separate rows: the 62 modes' brain-aging deltas, the 6 mode-clusters, the partialled versions of each, and the two separate all-in-one models of brain-age delta that use all 62 modes and all IDPs, respectively. The subscripts define whether the counts reported are the number of significant distinct SNP clusters for each phenotype, summed across modes/phenotypes ('SNPs'), or the number of modes/phenotypes with at least one association ('modes'). The superscripts describe the thresholding: either the standard single-GWAS threshold (7.5), the higher Bonferroni-adjusted threshold (9.33), or, in the case of the validation sample, the nominal 0.05 threshold (where here we are just reporting counts of validated associations from the higher discovery threshold).

| Phenotypes | Discovery | | | | Validation | |
|---|---|---|---|---|---|---|
| | $N_{SNPs}^{7.5}$ | $N_{SNPs}^{9.33}$ | $N_{modes}^{7.5}$ | $N_{modes}^{9.33}$ | $N_{SNPs}^{0.05}$ | $N_{modes}^{0.05}$ |
| 62 modes | 156 | 68 | 50 | 34 | 64 | 34 |
| 6 mode-clusters | 33 | 14 | 5 | 3 | 12 | 3 |
| 62 modes (partial) | 71 | 29 | 32 | 17 | 27 | 15 |
| 6 mode-clusters (partial) | 35 | 12 | 6 | 3 | 11 | 3 |
| all-in-one (62 modes) | 1 | 0 | 1 | 0 | 0 | 0 |
| all-in-one (IDPs) | 3 | 1 | 1 | 1 | 0 | 0 |

physical body measures (e.g. body size, fat, bone density variables and blood assays), cognitive test scores, and health outcome (including mental health) variables.

*Figure 3—figure supplements 2–3* show summarised results, and spreadsheets (Data availability) list every significant association. Below we describe many of these associations in more detail. In general, we focus on associations between partialled delta estimates and nIDPs, in order to identify associations specific to the unique brain-age-delta variance in modes.

## Individual modes: patterns of associations between the aging of the brain's structure and function and life factors, body measures, health outcomes and genetics

In *Figure 4* we list summary results of the strongest patterns of associations with brain-age delta from each mode-cluster and mode. We now expand on some of the more striking patterns in more detail.

Where a SNP discussed below is reported as an expression quantitative trait locus (eQTL) of a gene in the GTEx database (*Battle et al., 2017*), this means that variation in this SNP has been found to be highly correlated to variation in the gene expression. Many of the genetic associations described below passed the higher discovery threshold (as well as replicating), but we also discuss some associations that pass the lower (single-phenotype GWAS) threshold if they were also significant in the replication sample.

Mode-cluster $1_6$, which shows the strongest aging effect of the mode-clusters, is dominated by the volumes of the lateral ventricles and choroid plexus (and its intensity), the microstructure of the fornix and corpus callosum, and the volume of the thalamic nuclei. Notably, and of relevance to further results discussed below, fornix, choroid plexus and corpus callosum are all drained by the superior choroid vein, which runs along the whole length of the choroid plexus, and unites with the superior thalamostriate vein, which itself drains the thalamic nuclei (and striatum). Changes in diffusivity measures in the fornix, a thin tract in the immediate vicinity of the ventricles, may, however, be sensitive, indirect markers of the atrophy of the tract (resulting in a 'partial volume' reduction at voxel-level resolution), rather than representing a change to its white matter microstructure.

The non-imaging associations of mode-cluster $1_6$ included many modifiable risk factors such as heart rate, smoking, alcohol consumption and diabetes, (as well as taking metformin, a treatment for diabetes, although this is likely an indirect association that is essentially an indicator of the presence of diabetes). It is also associated with various measures related to overall non-fat body size: height,

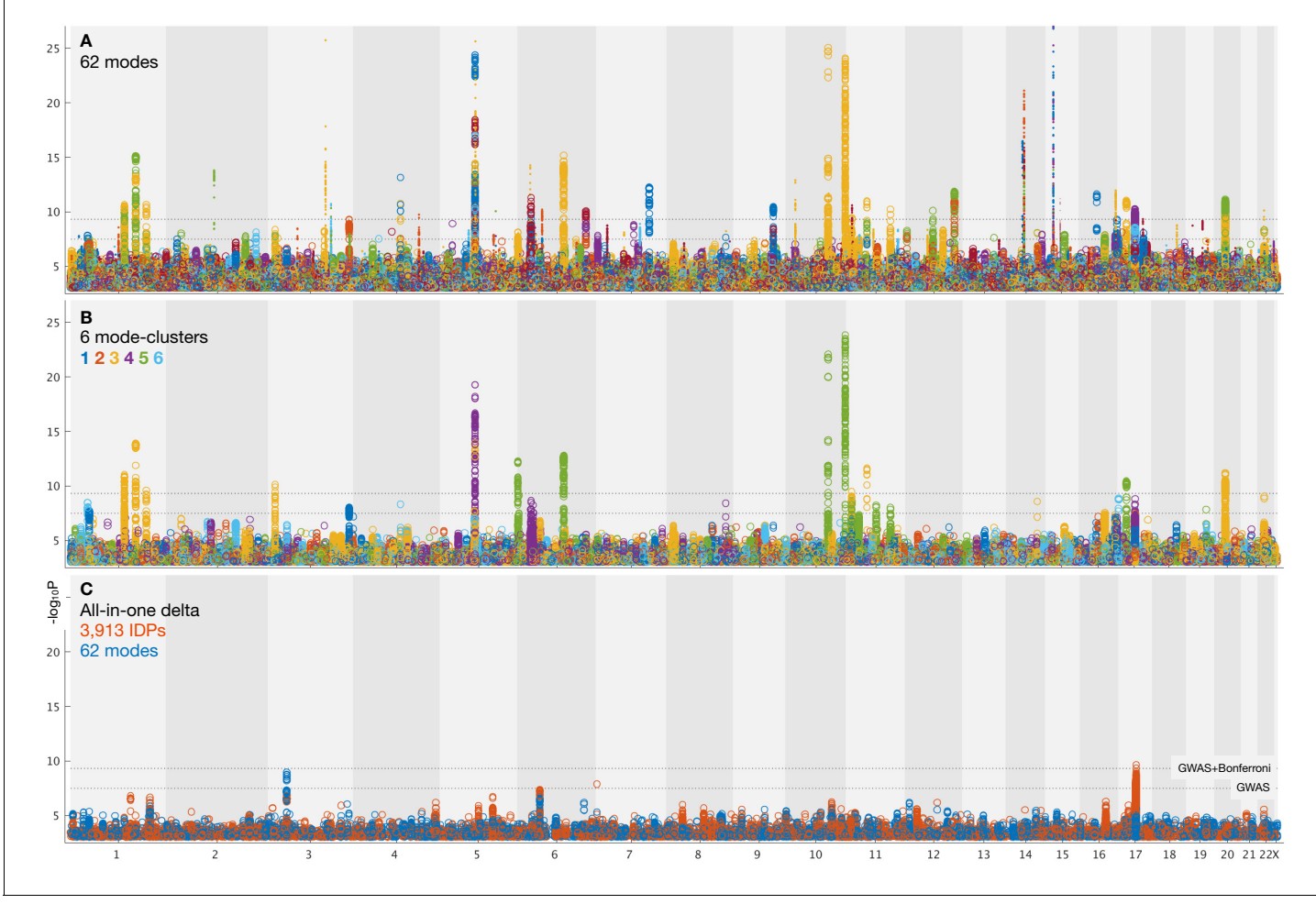

**Figure 3.** Summary plots for GWAS of brain aging. (**A**) Separate GWAS for each of the 62 modes of brain aging. The y axis is $-\mathrm{Log}_{10}P$ (significance of the genetic association) and the x axis is SNPs, arranged according to chromosomes 1:22 and X. For convenience of display some points of even higher significance (with redundant content compared with the points seen here) are truncated; for complete plots see *Figure 3—figure supplement 1*, and for individual plots (one per mode), see Data availability. The lower dotted line shows the standard GWAS threshold correcting for multiple comparisons ($-\mathrm{Log}_{10}P$ =7.5), and the upper line shows the result of an additional Bonferroni correction for the main 62+6 separate GWAS ($-\mathrm{Log}_{10}P$ =9.33). Circles denote the first 31 brain-aging modes (i.e., those with the strongest aging effect) and dots the next 31 (with weaker aging). (**B**) Separate GWAS for each of the 6 mode-clusters of brain aging. Again, see *Figure 3—figure supplement 1* and Data availability for complete and individual plots. (**C**) GWAS plots for two all-in-one estimates of brain-aging delta (with no points removed). In orange is shown the GWAS for the single delta estimated using all 3913 IDPs according to the approach in *Smith et al. (2019)*. In blue is shown the GWAS for the single delta estimated using the 62 modes. In both cases, the richness of genetic associations is clearly greatly reduced, compared with identifying distinct associations for each mode in its own right.

The online version of this article includes the following figure supplement(s) for figure 3:

**Figure supplement 1.** Summary plots for GWAS of brain aging.

**Figure supplement 2.** Mapping of brain-aging modes onto classes of IDPs, nIDPs and chromosomes.

**Figure supplement 3.** Mapping of brain-aging mode-clusters onto classes of IDPs, nIDPs and chromosomes.

strength, lung capacity, metabolic rate and weight, as well as multiple sclerosis. With regard to cognition, mode-cluster $1_6$ was associated with processing speed.

Consistent with the contribution of the identified modifiable risk factors, mode-cluster $1_6$ is associated with SNP rs4141741 (*MED8*), which was significantly correlated in the UK Biobank participants with blood pressure and diagnosed vascular and heart problems. The same SNP is an eQTL in the hippocampus of *TIE1*, which codes for a protein playing a critical role in angiogenesis and blood vessel stability, and of *MED8* in the striatum, both structures being innervated by the superior choroid and thalamostriate veins. Abnormal angiogenesis is also known to contribute to both diabetes and

| Mode-cluster | Mode-cluster IDPs | Mode-cluster nIDPs | Modes (IDPs and nIDPs) | Genes |
|---|---|---|---|---|
| $1_6$ | **CSF/ventricle volume** (both normalised for head size and not), fornix MD. | **Non-fat body size** (height, fat-free mass, lung capacity, grip strength, weight), metabolic rate, head bone density, cognitive speed, number of births. Diabetes, alcohol, smoking. | $2_{62}$ Fornix MD. <br> $11_{62}$ *CSF/ventricle volume*. Head bone area, TV time, cognitive score. <br> $12_{62}$ Thalamus volume. Body size. <br> $13_{62}$ Fornix MO. <br> $38_{62}$ Corona radiata FA. | *TIE1* <br> *MED8* <br> *GNA12* <br> *GMNC* |
| $2_6$ | **Grey volume**, white volume. | **Bone density** (primarily head), **non-fat body size** (height, fat-free mass, lung capacity, weight), metabolic rate, ankle width. Number of older siblings, cognitive speed. | $1_{62}$ Total grey volume. Bone density (doubled effect in females), age at menopause. Alcohol, smoking. <br> $16_{62}$ Amygdala/hippocampus volume. $19_{62}$ *Amygdala/hippocampus volume*. <br> $24_{62}$ *Medial-frontal cortical volume*. $25_{62}$ Superior-frontal cortical volume. <br> $32_{62}$ *Parietal/occipital cortical volume*. Maternal smoking, bone density, BMI. <br> $40_{62}$ Precuneus cortical volume. | *HRK* <br> *DAAM1* <br> *FAM3C* <br> *WNT16* <br> *CPED1* |
| $3_6$ | T1w cortical **grey-white contrast**. | BMI, weight, fat mass, haemoglobin. | $5_{62}$ Grey-white contrast, frontal. <br> $10_{62}$ Grey-white contrast, non-frontal. | *CREB3L4  CRTC2* <br> *SLC27A3  S100A16* <br> *STX6  WNT7A* <br> *CD82  VCAN* |
| $4_6$ | **MD in all white matter (particularly L2, L3).** | **Blood pressure / hypertension,** multiple sclerosis. | $4_{62}$ SLF MD, white matter lesions. <br> $6_{62}$ Superior corona radiata MD. Head bone area, smoking. <br> $7_{62}$ External capsule MD. Blood pressure. <br> $8_{62}$ Uncinate MD. $9_{62}$ *Posterior thalamic radiation / ILF MD*. <br> $14_{62}$ White matter lesions, posterior corona radiata MD. Multiple sclerosis. <br> $21_{62}$ Cerebral peduncle L2/L3. $22_{62}$ SLF MD. $27_{62}$ *Inferior cingulum MD*. <br> $30_{62}$ Superior cingulate gyrus: *L2/L3, MO/FA*. <br> $34_{62}$ Inferior cerebellar peduncle: *L2/L3, FA/MO*. <br> $52_{62}$ Putamen/caudate T2*, anterior internal capsule, anterior thalamic radiation ICVF. Haemoglobin, smoking, weight, meat intake. <br> $59_{62}$ Most white matter ISOVF/ICFV. Blood pressure, weight, smoking. | *VCAN* <br> *ZSCAN26* <br> *ZSCAN23* <br> *HLA-K* <br> *ZNF603P* |
| $5_6$ | **rfMRI amplitudes** (sensory, motor and cognitive). | **BMI, fat, weight, haemoglobin  red cell count, bone density, mobile phone use, income. Blood pressure, cardiac output, nervous feelings, sleep duration.** | $17_{62}$ Cerebellar/subcortical rfMRI amplitude. BMI, haemoglobin, weight, fat. <br> $31_{62}$ Cognitive cortex rfMRI amplitude. Physical activity, blood pressure treatment, fat. <br> $33_{62}$ Sensory/motor/cerebellar/subcortical rfMRI amplitude/connectivity. Heart rate, blood pressure, nervous feelings, night sleep duration, TV time, High SES / physical activity (mobile phone use, daytime sleeping, physical activity, drive fast, income, risk taking, number in house, number of sexual partners). <br> $41_{62}$ Visual rfMRI amplitude/connectivity. Age started wearing glasses. <br> $45_{62}$ *Sensory/motor rfMRI amplitude/connectivity*. Unenthusiastic, health-anxious, depressed, lack of physical activity. | *PLCE1* <br> *INPP5A* <br> *APOE* |
| $6_6$ | **Cortical thickness.** | BMI, weight, red cell count, head bone density. | $15_{62}$ Superior/medial frontal cortical thickness. <br> $23_{62}$ Precuneus/parietal. Birth weight. <br> $26_{62}$ Left lateral frontal. $28_{62}$ Right lateral frontal. <br> $29_{62}$ *Sensory/motor (central superior)*. <br> $39_{62}$ *Left post-central superior*. Number of older siblings. <br> $43_{62}$ Right post-central superior. | *MACF1* <br> *SLC39A8 / ZIP8* <br> *PAFAH1B1* |
| - | | | $3_{62}$ Fornix MD. Height, weight. $18_{62}$ Tapetum MD. $20_{62}$ Thalamus volume. Height, bone density. $35_{62}$ *Superior parietal cortex volume*. $36_{62}$ Putamen volume. TV time. <br> $37_{62}$ Subcortical T1 intensity. Weight, fat, nasal polyps. $42_{62}$ Cerebral peduncle / posterior internal capsule OD/FA/MO/L1. Body size, BMI. <br> $44_{62}$ Hippocampal/medial volume/cortical-area. $46_{62}$ Left Brodmann 44 cortical area/volume. $47_{62}$ Lateral orbital frontal cortical area/volume. Bone density. <br> $48_{62}$ *Right (mostly lateral) occipital area/volume*. Maternal smoking. $49_{62}$ *Corticospinal tract MD*. BMI, snoring, Body size. <br> $50_{62}$ *Superior cerebellar peduncle MD*. Height, IQ, number of older siblings, TV time, driving time. $51_{62}$ *Right Brodmann 44 cortical area/vol*. <br> $53_{62}$ *Posterior thalamic radiation FA/MO L2/OD*. Weight, fat, number of older siblings, Glaucoma. $54_{62}$ Right Brodmann 45 cortical area/volume. <br> $55_{62}$ *Left Brodmann 45 cortical area/volume*. $56_{62}$ Cuneus volume. $57_{62}$ Left (mostly lateral) occipital area/volume. <br> $58_{62}$ Calcarine/lingual area/volume. Maternal smoking. $60_{62}$ *Inferior temporal area/volume*. $61_{62}$ Parahippocampal/entorhinal volume. $62_{62}$ *Cingulate volume/area*. | |

**Figure 4.** Dominant imaging, non-imaging and genetic associations between brain-age delta from all mode-clusters and modes. The left side of the table focuses on the main patterns of associations with the 6 mode-clusters, while the right side also lists dominant associations with individual modes, grouped according to the mode-clusters. At the bottom are results from individual modes that do not have one clear associated mode-cluster. Red text signifies positive correlation with brain-age delta (meaning in general a detrimental factor with respect to aging), and blue indicates negative correlation (i.e. a positive causal factor and/or outcome with respect to aging). Where the all-in-one brain-age modelling has negative $\beta$, the signs of associations between delta and IDPs becomes the inverse of the original ICA IDP weight; in such cases, the table makes this appropriate adjustment to text colour (such that the colour reflects the sign of assocation between delta and IDP, and not ICA weight), but we denote where this occurs by use of italics. Bold text indicates relatively stronger associations (in terms of strength of effects and/or number of related variables). Results included here are generally stronger than $-\text{Log}_{10}P$ >7 for nIDPs (see Materials and methods), and SNPs are listed only where replication succeeded. To help focus the descriptions of non-imaging variables, we largely list their associations with the partialled deltas; this therefore concentrates on unique variance in deltas. When working with partialled variables (or equivalently multiple regression), and when adjusting for some of the imaging confounds (such as head size, when considering volumetric measures), signs of associations can in some cases be non-trivial to interpret.

The online version of this article includes the following source data for figure 4:

**Source data 1.** Spreadsheet version of *Figure 4*.

multiple sclerosis, perhaps explaining to some extent our non-imaging association results with both these diseases.

Modes related to mode-cluster $1_6$ include $2_{62}$ and $11_{62}$. Mode $11_{62}$ (ventricle volume) is associated with SNP 7:2777917_TA_T (rs1392800372), which is likely in gene *GNA12*; this has been found to relate to migration of neurons in the developing brain (*Moers et al., 2008*). This may therefore be relevant in the context of the neural stem cell pool in the subventricular zone (*Ruddy et al., 2019*), that is, relating these modes to ventricle size and neuronal development/angiogenesis. In line with mode-cluster $1_6$ being dominated by the volumetric measure of CSF (cerebro-spinal fluid, which fills the ventricles), mode $2_{62}$ (fornix MD) is associated with SNP rs150434736 (on chromosome 3, only 17kbp from the 3:190657741_AGT_A/rs147817028 peak in mode-cluster $1_6$), near gene *GMNC*; this has been found to be linked to Alzheimer's disease endophenotypes (in particular ptau 181 in CSF) (*Cruchaga et al., 2013*; *Deming et al., 2017*).

Mode-cluster $2_6$ relates to global measures of grey and white matter volume. It was associated with body-size-related non-imaging measures in common with mode-cluster $1_6$, including those of height, weight, strength, metabolic rate and lung function, and also cognitive reaction time.

Several related modes (in particular, $16_{62}$, $19_{62}$, $24_{62}$, $25_{62}$, $32_{62}$ and $40_{62}$) relate to regional (i.e. more focal) grey matter volume. These modes did not have many nIDP associations (i.e. the nIDP associations for the mode-cluster were largely not regionally specific to individual modes), although mode $32_{62}$ (parietal/occipital volume) was associated with maternal smoking. While mode-cluster $2_6$ as a whole did not have strong genetic associations, some of these regional-grey-volume modes did. Modes $16_{62}$ and $19_{62}$ (hippocampus volume) were found to be significantly associated with *HRK*; this is involved in apoptosis/neurogenesis, particularly in adults in hippocampus (*Coultas et al., 2007*), and expressed (eQTL) in hippocampus. Mode $40_{62}$ (volume of the precuneus cortical region) was associated with *DAAM1* (*Elliott, 2018*; *Mollink et al., 2019*), important for cell polarity and neural development.

While the above modes relate to regionally-specific grey matter volume, there is also involvement (in this mode-cluster) of mode $1_{62}$; this codes for total grey matter volume and is the most strongly age-related mode. This mode is associated with smoking and alcohol, as well as bone density (as measured separately from the MRI, using DEXA low-dose x-ray and also ultrasound). This bone density association is strong, reaching $r = 0.43$ in females and 0.27 in males. The greater bone density loss in females is likely to be associated with menopause. Firstly, this mode is significantly associated with age-at-menopause (a non-imaging variable in UK Biobank, with average age-at-menopause being 50y). More generally, there is a large amount of literature showing that bone density loss is specifically accelerated in the 10 years after menopause (*O'Flaherty, 2000*); this exactly matches the sex-specific pattern of change in females seen in this mode (*Figure 5F,G*).

*Figure 5A, B* shows the increase in T1-weighted intensity within the skull, associated with this mode. This is reflecting an increase in bone marrow fat with increasing brain-age delta. This, together with the above nIDP associations with bone density loss in this mode, is consistent with literature regarding decreasing bone density and increasing marrow in aging (*Cordes et al., 2016*). Bone density reduction has not just been reported in normal aging, but has also been linked to early Alzheimer's disease, independent of age, sex, habitual physical activity, smoking, depression and estrogen replacement status (*Loskutova et al., 2009*).

These results are consistent with the one strong genetic association with mode $1_{62}$ (*Figure 5E*); lead SNP rs3801383 (whose PHEWAS results are dominated by bone density associations - see Materials and methods) lies within the span of *FAM3C*, but also is in LD with SNPs spanning across to genes *WNT16* and *CPED1* (*Chesi et al., 2015*; *Movérare-Skrtic et al., 2015*). *FAM3C* is associated (in UK Biobank genetic data http://big.stats.ox.ac.uk) with bone density loss and bone fractures, but has also been linked directly to Alzheimer's disease through impact on brain amyloid (*Liu et al., 2016*).

Mode-cluster $3_6$ singles out IDPs representing T1-weighted intensity contrast between white and grey matter (across the grey-white border). Although mode-clusters $1_6$ and $2_6$ related to weight, they were essentially driven by non-fat mass; here, however, mode-cluster $3_6$ mainly relates to measures of fat mass and fat percentages across the body, as well as blood haemoglobin measures.

In line with these non-imaging correlations, one strong genetic association was found for a SNP (rs12133923), an eQTL of *CREB3L4* in the basal ganglia and of *CRTC2*, *SLC27A3* and *S100A16* in the cerebellum. *CREB3L4*, which regulates adipogenesis, has for instance been recently shown to have a

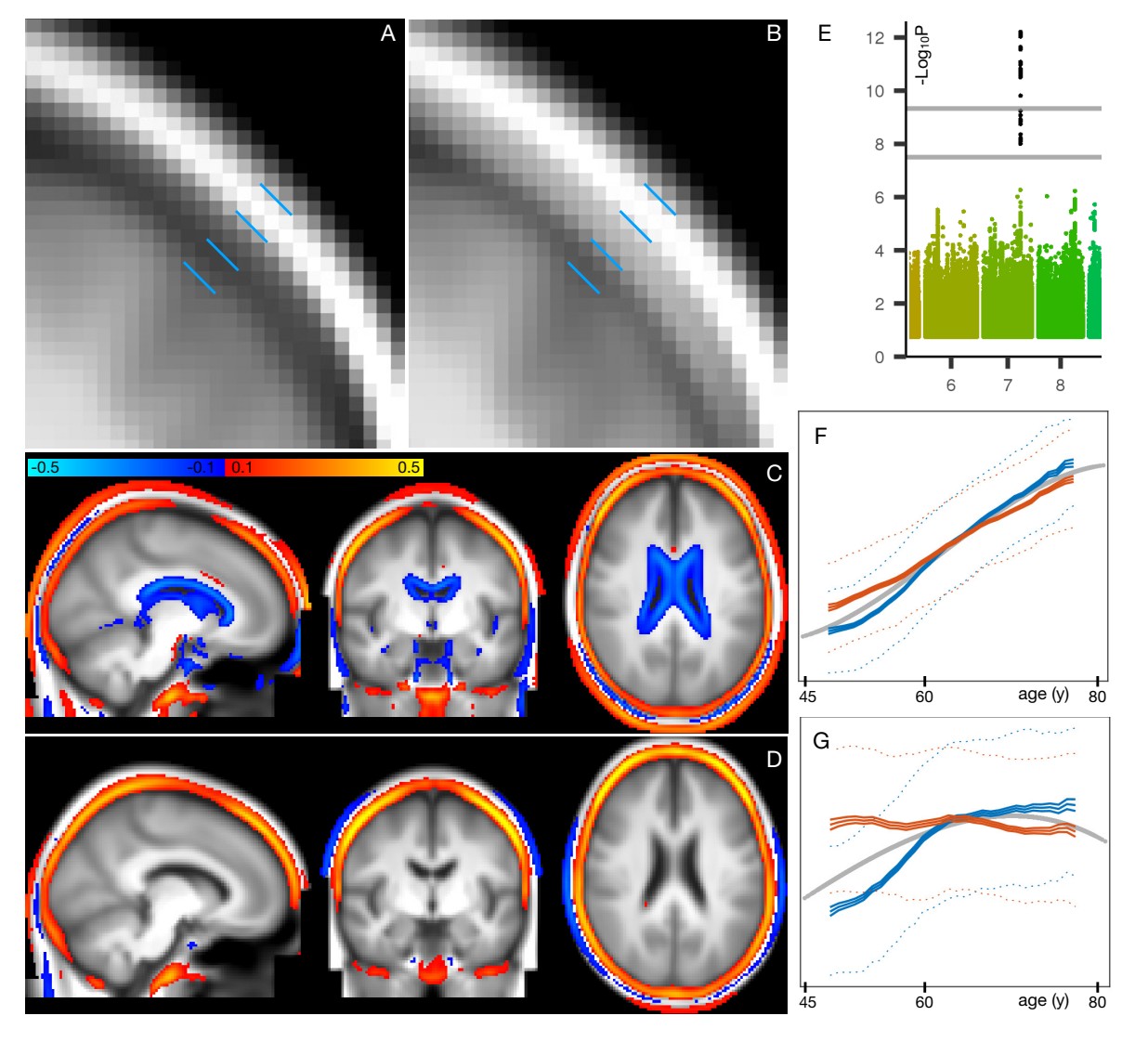

**Figure 5.** Spatial mapping of mode $1_{62}$ onto original T1-weighted MRI data, along with genetic and age-dependent plots. (**A**) A region-of-interest from the average T1-weighted structural image from the 1000 subjects with the lowest delta values for this mode. The images have been linearly-aligned into standard (MNI152) template space, and have not been brain-extracted, so that non-brain tissues can be seen. The blue lines delimit 3 'layers' seen in cross-section; from the outside in, these are skin/fat outside the skull, the skull, and cerebrospinal fluid outside of the brain. (**B**) The equivalent average image from the 1000 subjects with the highest delta values. There is no obvious geometric shift (e.g. of tissue boundaries), but the intensity values are clearly higher within the skull; this is reflecting increase in bone marrow fat with brain-age delta. (**C**) The difference between the two average images (all images were first normalised to have a mean intensity of 1). (**D**) The same difference of averages, but after regressing all confounds (including age) out of the voxelwise imaging data, and working with the partialled delta values for mode $1_{62}$; with this more focussed analysis, changes around the ventricle are no longer obvious, but the change in skull intensity remains. (**E**) The one significant genetic association (on chromosome 7) for this mode. The lower grey line shows the standard single-phenotype threshold of 7.5; the upper line shows this after Bonferroni adjustment for multiple tests (modes). This significant association was also found in the replication dataset. (**F**) The mean age curves for mode $1_{62}$ (as described in more detail in Materials and methods and *Figure 1—figure supplements 3–9*). Females are shown in blue, males in orange; the y axis is the unitless mode subject-weights (averaged across subjects with an averaging sliding window). The greatest rate of age-related change is in females, in the 10y following menopause. (**G**) This pattern is even more striking in the partialled subject-weight curves (where other modes have first been regressed out of mode $1_{62}$.).

critical role in metabolic phenotypes (weight gain, impaired glucose tolerance and decreased insulin sensitivity) (*Kim et al., 2015*). *CRTC2* plays a role in lipid metabolism, and *SLC27A3*, which encodes fatty acid transport protein, is involved in the developmental stage of the central nervous system (*Maekawa et al., 2015*). Taken altogether, it is therefore likely that the marked, widespread change

of cortical contrast with aging witnessed here (and in several previous studies) is strongly related to the fatty, lipid-rich myelin (*Salat et al., 2009*; *Vidal-Piñeiro et al., 2016*; *Lewis et al., 2018*).

Of note, mode-cluster $3_6$ was also strongly associated with other SNPs, amongst them one (rs1044595) in an exon of *STX6*, and in high LD with a SNP associated with tauopathy progressive supranuclear palsy (*Höglinger et al., 2011*), and correlated in the UK Biobank participants with hormonal replacement treatment. Another hit (rs6442411), an eQTL of *WNT7A*, which regulates angiogenesis, neurogenesis and axon morphogenesis, was associated in the UK Biobank population with height and trunk mass. One SNP, rs541397865, was found in an intron of *CD82*; this regulates the migration of oligodendrocytes, which are responsible for axonal myelination. We also found a genetic association with rs10052710, a SNP in an intron of *VCAN*, and in high LD with a previous hit we had found strongly associated with diffusion measures across the entire white matter *Elliott (2018)*. These additional associations further point to the driving contribution of myelin in the aging-related modulation of grey/white-matter contrast.

Mode-cluster $\mathbf{4_6}$ is strongly linked to modifiable risk factors: high blood pressure, vascular and heart problems, and associated with taking ramipril (a treatment against high blood pressure and heart failure). It was also associated with a number of illnesses and treatments, including multiple sclerosis. Mode-cluster $4_6$ is characterised by diffusion measures of mostly frontal white matter (anterior corona radiata and, overlapping in the frontal lobe, the inferior fronto-occipital fasciculus), and was also associated with general reaction time. The subject-weights are strongly age-dependent (as are all mode-clusters); however, they have very little age dependence after partialling out other mode-clusters; this means that the above factors interact in a manner that is largely age-independent.

Genetic associations included again a SNP in an intron of *VCAN* rs17205972, in high LD with the *VCAN* SNP associated with mode-cluster $3_6$, and reported in *Elliott (2018)*. Additionally, there was association with SNP rs3129787, an eQTL in the brain of *ZSCAN26* and *ZSCAN23* (in the cortex and cerebellum), *HLA-K* (cortex), and *ZNF603P* (basal ganglia, cortex, hypothalamus, cerebellum), a pseudogene whose expression in the brain has been recently observed to be associated with schizophrenia and affective disorders (*Bhalala et al., 2018*). The latter SNP was also highly correlated in the UK Biobank participants with health issues including coeliac/malabsorption disease, blood pressure, taking insulin and hyperthyroidism, as well as with measures of lung function.

Mode-cluster $\mathbf{5_6}$ shows a modest deceleration of aging-rate with increasing age, particularly with respect to its unique (partialled) variance (*Figure 1—figure supplement 9*). It involves just the amplitudes of resting-state fluctuations, covering most of the brain; some of the associated modes also show rfMRI connection-strength involvement, but that may be an indirect result of the amplitude changes. Mode-cluster $5_6$ demonstrated strong correlations with non-imaging variables similar to mode-cluster $3_6$: weight, fat mass and percentage, red blood cell count and haemoglobin. It also was associated with blood pressure, cardiac output and bone density, along with sleep duration, nervous feelings and several markers of socio-economic status (SES).

Mode-cluster five was strongly associated with several SNP clusters, having relevant correlations in the UK Biobank population: rs7766042 with snoring; rs2273622 with high blood pressure, migraine and headache, taking pain relief, vascular and heart problems; and rs2274224 with weight, (fat-free, and fat) mass, fat percentage and blood pressure (including taking amlodipine). This latter SNP is in an exon (missense) of *PLCE1*, as also seen in *Elliott (2018)* and (*Hübel et al., 2019*), another recent UK Biobank study on body fat percentage. The strongest GWAS hit is rs4497325, and for the associated mode $45_{62}$, the peak SNP is (the immediately-neighbouring) rs7096828; this is an eQTL of *INPP5A*, which is involved in DNA methylation in neurons, associated with aging and depression (*Gasparoni et al., 2018*).

Finally, a genetic association was found with rs429358, the SNP that determines whether the APOE allele is ε4 or not. This is a major locus associated with Alzheimer's disease and mild cognitive impairment, and also with dementia with Lewy bodies, age at onset of symptoms in Parkinson's disease, insomnia, brain amyloid deposition and neurofibrillary tangles, inflammation, HDL/LDL cholesterol and triglycerides levels, physical activity and blood protein levels, parental longevity, and macular degeneration. In the UK Biobank participants http://big.stats.ox.ac.uk, this SNP also correlated with Alzheimer's disease in father/mother/siblings, LDL/cholesterol levels (and taking cholesterol-lowering medication), omega6, triglycerides, diabetes in the mother, weight and fat mass, with heart disease and with the mother's and father's age at death, amongst many other variables.

Despite being associated with SES, mental health markers, functional MRI amplitude fluctuations, and SNPs involved in cognitive decline, there were no direct associations between this mode-cluster (or its associated modes) and cognitive test scores. In the case of the IDPs, this may well mean that the changes seen are non-neural effects (e.g. cardiovascular causes of changes in the BOLD amplitude), and that any associated cognitive effects are caused by ongoing damage and not seen until later in life than the majority of the samples (imaged subjects) here. Even in the mode covering cognitive brain regions ($31_{62}$), the set of nIDPs is dominated by exercise/activity measures and not cognitive test scores. The link between SES and fMRI activity levels seen in Figure 7C in *Miller et al. (2016)* may now be explained; here, mode $41_{62}$ links amplitude of rfMRI fluctuations in visual cortex to age when started using glasses (and indeed we looked at age subgroups to confirm that this association is driven by those subjects who started wearing glasses while younger than 30y).

Mode-cluster$6_6$ was entirely composed of grey matter thickness IDPs, mainly in the prefrontal areas, as well as higher order parietal and temporal regions. It correlated with non-imaging variables of weight, red blood cells and head bone density. This mode-cluster was age-dependent, but its unique (partialled) variance was only weakly so.

We found three genetic associations with Cluster 6. The first, rs682571, is in an intron of *MACF1*, which has been shown recently to regulate the migration of pyramidal neurons and cortical GABAergic interneurons (*Ka et al., 2014*; *Ka et al., 2017*). This SNP also correlated in the UK Biobank population with several measures of body fat. Another hit, rs13107325, is in an exon (missense) of *SLC39A8* (*ZIP8*), the same SNP reported in our GWAS-IDP study (*Elliott, 2018*) to be associated with subcortical and cerebellar volume and susceptibility. This has also been found in other GWAS studies (many based primarily on UK Biobank data), including those looking at medication use, tobacco and alcohol consumption, cholesterol, body fat, adiposity, osteoarthritis, red blood cell, blood pressure, sleep duration, risk taking, intelligence/math ability/cognitive function and schizophrenia. A final SNP, rs7219015, was found in an intron of *PAFAH1B1* that, when mutated, leads to lissencephaly. It is also found to correlate with tiredness in UK Biobank (*Deary et al., 2018*).

## Discussion

Here, we aimed to study how multiple, biologically distinct, modes of population variation in brain structure and function reflect different aspects of the aging brain. We investigated the modes' distinct associations with genetics, life factors and biological body measures, in the context of the modelling of brain age and brain-age delta - a measure of whether subjects' brains appear to be aging faster or slower than the population average.

To study these multiple modes, we used brain imaging data from six different imaging modalities spanning many different aspects of brain structure and function, from 21,407 subjects, from a single, highly homogeneous, study. All imaging data were first reduced to 3,913 IDPs (imaging-derived phenotypes - summary measures of brain structure and function) from across the different modalities. However, rather than studying aging in different individual IDPs, we identified latent factors of population covariation using unsupervised learning, to provide a more compact, lower-noise representation of the population data, and focussing only on population modes showing extremely high split-half reproducibility.

All imaging data (and the same set of IDPs) used for our work here are available from UK Biobank, as is all code used for the core UK Biobank processing, and new code generated for this work is also freely available. Therefore, for data from other (non-UK Biobank) studies, the full code is available for deriving the exact same set of IDPs, as long as the same imaging modalities are acquired. How well harmonised those IDPs would be with UK Biobank IDPs would of course be a 'sliding-scale', dependant on how similar the MRI scanner hardware, scanner software and protocol were to those in UK Biobank. Similarly, how similar any derived brain modes would be to those that we report here would likewise be a sliding-scale, dependant on how similar the data characteristics (and subject group demographics) were.

Previous work showing more than a single pattern of brain aging includes (*Groves et al., 2012*), where we used voxel-level multimodal independent component analysis (ICA) applied to data from 484 subjects, to generate multiple population modes, several of which showed age dependence (including early-life development). However, this data spanned almost the entire human age range (8-85y), with data from just two imaging modalities, and hence did not identify a large number of

distinct modes relating to older-age aging. In the same year, a study of early-life development and maturation (885 subjects, 3-20y) used three imaging modalities to generate 231 distinct imaging features (*Brown et al., 2012*). The features were then grouped into different subsets by hand, and the age dependence of each subset (and also of many of the features on their own) was studied. Similarly, (*Vinke et al., 2018*) included data from several modalities, and studied aging trajectories in different measures from different modalities, but did not go as far as brain age (or brain-age delta) modelling, or attempt to identify latent modes of aging. Several modalities were also used in *Richard et al. (2018)*, with 11 groups of distinct measures used to form 11 estimates of brain age, each of which was then separately investigated for cognitive associations; one central methodological distinction to the work presented here is that the 11 models were hand curated according to different types of features from different modalities, as opposed to (in our case) pooling all modalities' features together before using data-driven decomposition (ICA) to identify distinct aging modes that could naturally span across feature types and modalities. In contrast, *Kessler et al. (2016)* used single-modality features (resting fMRI edge strengths) fed into ICA to identify multiple modes of early-life maturation. Finally, *Kaufmann et al. (2019)* used a single imaging modality (T1-weighted structural images) from 45,000 subjects pooled from 40 studies, to investigate the relationship between brain aging and several diseases. Brain-age prediction was trained from whole-brain analysis of the structural data, and also seven atlas-defined regional subsets were used to retrain the predictions. The different regional brain-age delta estimates showed varying associations with disease. However, as with our all-in-one predictions and also (*Ning et al., 2018*), direct GWAS of the delta estimates showed virtually no significant assocation, even with these high subject numbers.

We suggest that there is value in considering multiple, multimodal, brain aging modes separately; for example, while our single all-in-one modelling of brain-age delta had no significant genetic influence, many of the individual modes had significant, rich and biologically interpretable genetic influence. We also found rich patterns of significant associations with non-imaging non-genetic variables, including: biological measures (bone density, body size and fat measures, metabolic and cardiovascular function, blood pressure, haemoglobin, age at menopause); life factors (alcohol, smoking, maternal smoking, physical activity, number of siblings, sleep duration, many markers of socio-economic status); cognitive test scores (processing speed, IQ); mental health (anxiety, depression); and disease (diabetes, multiple sclerosis). To help focus our reporting of these non-imaging variables, we largely considered their associations with the partialled deltas, i.e., concentrating on unique variance in each mode's delta. However, doing this is not mandated where the non-imaging variables (e.g. blood pressure) or genetics are more likely to be causal factors than caused, in which cases, the (in general less conservative) correlations with non-partialled deltas can be more appropriate.

The multiple modes of brain aging involved all imaging modalities, in a range of different patterns. Some modes spanned multiple modalities, while others were more focussed, primarily reflecting within-modality patterns. Measures of brain structure and function included: volumes of grey and white tissues and structures (e.g. ventricles, thalamus, hippocampus); intensity contrast between grey and white matter; microstructural measures in white matter tracts (diffusivity, anisotropy); amplitude of spontaneous fluctuations in grey matter fMRI amplitude, and functional connectivity between regions; volume of lesions in white matter; and changes in susceptibility-weighted contrast (likely reflecting iron deposition) in subcortical structures. Of course, while all imaging modalities did show some involvement in the brain-aging modes, it is not the case that all are equally valuable, both in terms of reflecting the true underlying biology of brain aging, or in terms of the reliability with which they could be estimated from the UK Biobank imaging data. For example, the fMRI IDPs in general seem noisier than structural IDPs, as seen in our previous GWAS results (*Elliott, 2018*); however, it would be incorrect to conclude that the noisier IDPs contain no useful information, and indeed we showed in the same study that a data-driven reduction of hundreds of individual fMRI IDPs to a small number of latent factors did show significant genetic association. Similarly, here, the fMRI-based modes were not strongly dominant, but were nevertheless reproducible (see Materials and methods). Additionally, having included the 'relatively noisy' fMRI IDPs did not damage estimation of more structural modes: if we exclude fMRI IDPs from all analysis, and rerun the mode-cluster estimation, the five mode-clusters not dominated by fMRI IDPs were still found (in all cases having high correlation, $r>0.9$, with their original estimation).

Although there is a good deal of literature relating patterns of normal brain aging to some diseases (including our results discussed above), one should not assume that all diseases display

patterns identical to accelerated normal brain aging. This does not mean that the study of normal brain aging would not be of value in such diseases; indeed, thorough characterisation of normal brain aging could well help disentangle disease effects from (non-disease) aging effects in the subjects with disease. Additionally, identification of latent factors of population variation (such as carried out here) may help in the discovery of distinct disease sub-groups.

All modes' subject-weight vectors are oriented (by definition) to increase with age. However, with respect to their unique signal (obtained by regressing out all other modes from any given mode), a small number of modes are negatively correlated with age, as described in Materials and methods and shown in *Figure 1—figure supplement 2*. Two examples are visualised in *Figure 6*. Mode $53_{62}$ involves changes in white matter fibre organisation in the posterior thalamic radiation (also known as the optic radiation, and connecting to visual cortex), and was associated with glaucoma, as previously reported (*Wang et al., 2018*). Mode $50_{62}$ involves changes in white matter diffusivity in the superior cerebellar peduncle, and was associated with IQ and several markers of socio-economic status. In such cases, where a mode's unique variance is contributing to reducing (and not increasing) brain age, one possible interpretation is that the mode represents cognitive reserve, that is, working against the general pattern of age-related decline (indeed, aspects of socio-economic status are frequently used as proxies for cognitive reserve). The primary goal of this work is the decomposition of aging effects in the brain into multiple modes, which would ideally be biologically distinct and interpretable. It is unsurprising that, by incorporating all modes (and hence all IDPs), the all-in-one brain-age modelling achieves higher accuracy in age prediction; however, this is achieved at the expense of diluting associations, as seen here with the genetics. It nevertheless could be the case that deeper consideration of how the modes interact with each other to achieve optimal integrated modelling (for example, considering their 'partialled' regression parameters in depth) may bring new understanding about brain aging. For example, this could shed new light on how different external causal factors and distinct brain aging responses to these interact with each other.

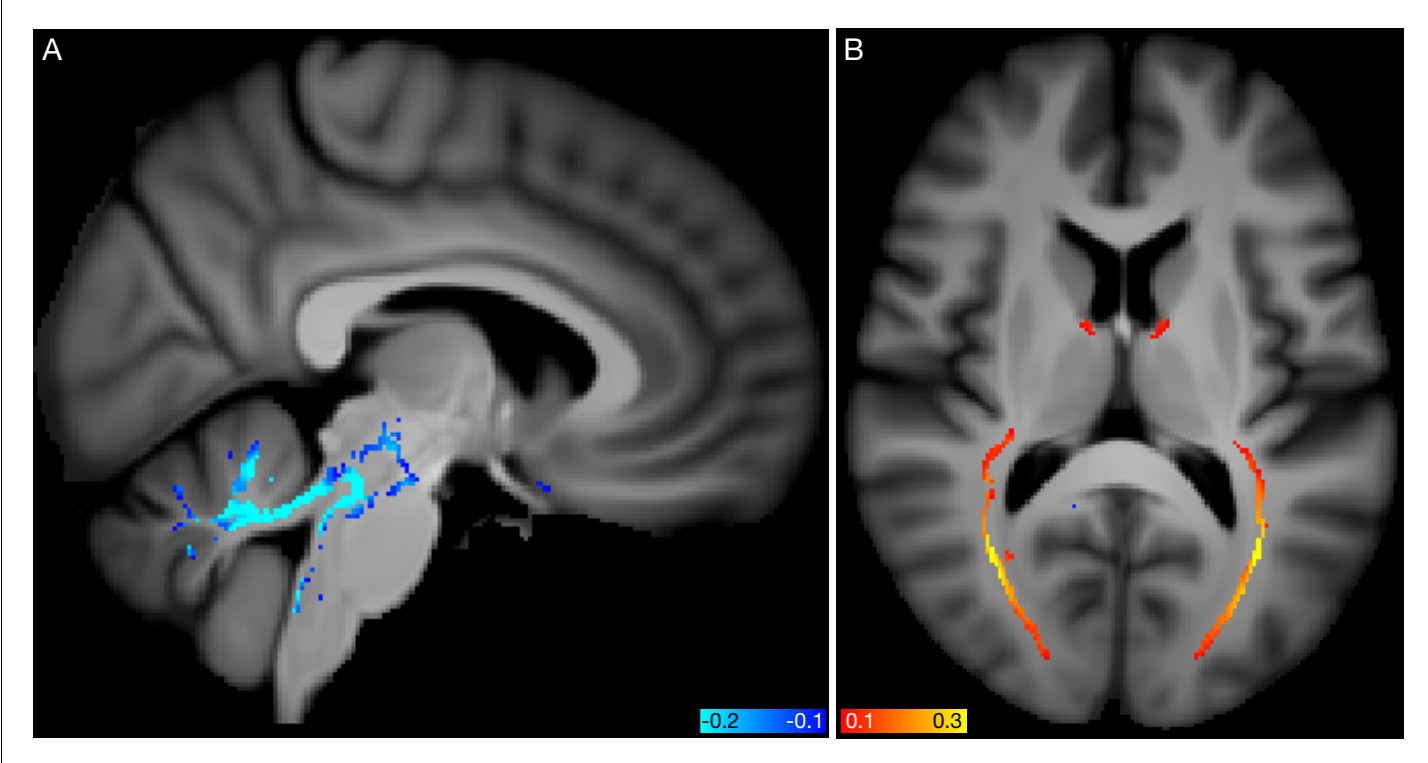

**Figure 6.** Spatial mapping of modes $50_{62}$ and $53_{62}$ from the diffusion MRI data. (A) Voxelwise correlation of the partialled brain-age delta values (one per subject) from mode $50_{62}$, into the dMRI MD (mean diffusivity) data. The colour overlay shows correlation *r* values, thresholded at a magnitude of 0.1. (B) Voxelwise correlation of the partialled brain-age delta from mode $53_{62}$, into the dMRI FA (fractional anisotropy) data.

A major premise behind the modelling of multiple brain-aging modes is that each mode in each subject has a distinct delta: for a given subject, the different modes are 'aging' differently. Although the common approach in the brain-aging literature is to estimate a delta (or brain age gap) by subtracting actual age from estimated brain age, this has the potential weakness of assuming that this offset would be constant for a given subject, as the subject gets older. For example, this assumption is implicit when looking for genetic associations, as one would like to be finding associations with an age-independent marker of relative brain health. However, it may be more likely that (for example) a given subject's brain is aging faster than the population average in terms of a distinct aging rate, implying that their delta would be increasing over time (and not therefore being a constant offset relative to their age). Indeed, our results show that there is evidence for this being a more appropriate model of brain aging (*Figure 1—figure supplement 10* shows several modes where the variance of delta increases with age, and a few where it decreases). Unfortunately, the two models can be hard to distinguish, particularly at the level of individual subjects, when given only single-time-point (cross-sectional) data. For example, it may be hard to disambiguate whether total brain volume is different than the population average because of aging effects, or because the subject had a larger/ smaller brain at 'baseline' (before age-related decline began). This is similar to the distinction between 'shallow' vs. 'lagged' early-life maturation investigated by *Kessler et al. (2016)*. Naturally some preprocessing helps ameliorate this, for example, normalisation of brain volume by head size. However, it is still the case that longitudinal data, and more advanced modelling, may result in more sensitive and meaningful study of brain aging in future. Notably, UK Biobank has now started re-imaging 10,000 of the 100,000 subjects, with an average scan-rescan interval of about 2 years. Raw and preprocessed data from almost 1500 of these rescanned subjects will be released in early 2020. Future work on brain-age modelling can hope to take advantage of the ever-increasing size and richness of such datasets, to enable better understanding of the aging brain in health and disease.

## Materials and methods

### Data and preprocessing

We used data from 21,407 participants in UK Biobank, 53% female, aged 40-69y at time of recruitment and 45-80y at time of imaging. As described in detail in *Miller et al. (2016)*, the UK Biobank data includes 6 MRI modalities: T1-weighted and T2-weighted-FLAIR (Fluid-Attenuated Inversion Recovery) structural images, susceptibility-weighted MRI (swMRI), diffusion MRI (dMRI), task functional MRI (tfMRI) and resting-state functional MRI (rfMRI).

We (and colleagues) have developed and applied an automated image processing pipeline on behalf of UK Biobank (*Alfaro-Almagro et al., 2018*) https://www.fmrib.ox.ac.uk/ukbiobank/fbp. This removes artefacts and renders images comparable across modalities and participants; it also generates thousands of image-derived phenotypes (IDPs), distinct measures of brain structure and function. Here we used 3913 IDPs available from UK Biobank, spanning a range of structural, diffusion and fMRI summary measures (as described in the central UK Biobank brain imaging documentation http://biobank.ctsu.ox.ac.uk/showcase/showcase/docs/brain_mri.pdf and listed in full in a spreadsheet available at https://www.fmrib.ox.ac.uk/ukbiobank/BrainAgingModes).

Code for all processing in this paper is freely available (see Data availability). Each IDP's $N_{subjects} \times 1$ data vector had outliers removed (set to missing, with outliers determined by being greater than 6 times the median absolute deviation from the median); the vector was then quantile normalised [*Miller et al., 2016*], resulting in each IDP's data vector being Gaussian-distributed, with mean zero, standard deviation one. We then discarded subjects where 50 or more IDPs were missing (for any reason, which could be due to: data acquisition incompleteness; data quality problems as described in *Alfaro-Almagro et al., 2018*; or the above-described outlier removal), leaving 18,707 subjects (54% female). The small amounts of remaining missing data were replaced with close-to-zero values (random signal of standard-deviation 0.01). This resulted in an IDP data matrix $W$ of size 18,707×3,913.

Confounds were removed from the data as carried out in *Elliott (2018)* (except that age-dependent confounds were not removed from $W$). This includes confounds for: head size, sex, head motion during functional MRI, scanner table position, imaging centre and scan-date-related slow drifts.

In applications with a specific disease of focus, it is common to generate a model such as brain-age-estimation based on healthy subjects and then apply it to both healthy and disease subjects. However, here (and in UK Biobank in general) there is no one specific disease focus, with all diseases being of potential interest, and with the imaged population being largely healthy at the time of imaging. The fractions of imaged subjects having specific existent diagnoses are low (e.g. with less than 10% having mental health or neurological diagnoses, and none having gross anatomical pathology according to the processing pipeline QC *Alfaro-Almagro et al., 2018*). We therefore did not exclude individual subjects from the modelling here.

## Estimation of multiple population modes of brain aging

We then applied independent component analysis (ICA), using the FastICA algorithm (*Hyvärinen, 1999*). ICA decomposes a data matrix into multiple factors that are statistically independent of each other with respect to one of the data matrix dimensions (the input data matrix is $W$, meaning that the data dimensions are subjects and IDPs). This generates multiple independent modes of population covariance (patterns of IDPs that co-vary together across subjects).

In order to help focus this data-driven decomposition on age-related population modes, both with respect to the pre-ICA dimensionality reduction (achieved using PCA - principal component analysis) and the core ICA unmixing, each IDP vector $W_i$ (after normalisation as described above) was rescaled by an age-related factor of $(0.1 + abs(corr(age, W_i)))$, before PCA+ICA was applied.

ICA requires the estimated (output) components to be non-Gaussian in their distributions, and our data matrix W is more highly non-Gaussian in the IDP dimension than in the subject dimension (which is largely Gaussian for most IDPs, even before quantile-normalisation). We therefore appply ICA to estimate modes of independent IDP weights. Each ICA component therefore comprises a mode of population covariation described by two vectors: the 'ICA source vector', spanning all IDPs, with one (signed) scalar weight value per IDP; and the subject-weights vector, with one (signed) scalar weight value per subject. The rank-1 outer product of these two vectors comprises this mode's contribution to the full original data matrix. IDP-weight-vectors are statistically independent of each other (by definition, according to the ICA algorithm) and hence also orthogonal, whereas the subject-weight-vectors are only restricted to being non-co-linear (and indeed below we utilise their correlations with each other to help identify clusters of modes).

Estimation of association of a given mode with age or non-imaging variables (such as cognitive test scores and physical body measures) can proceed simply by correlating/regressing the subject-weights vector against any relevant non-imaging $N_{subjects} \times 1$ vector. As described above, all modes have distinct (from each other) subject-weights-vectors and IDP-weights-vectors, and hence are distinct modes of population variation. Note that the ICA algorithm will always produce the requested number of modes, and as such the statistical robustness of identified modes requires some form of test, such as the reproducibility testing described below.

A major controlling parameter in an ICA decomposition of a data matrix is the number of components it is asked to estimate - that is, how fine-grained the 'clustering' output should be. It is common to specify just one controlling parameter when running FastICA, that being the initial PCA dimensionality reduction. ICA would then output the same number of components. However, it is also possible to control the PCA dimensionality, and separately determine which ICA output components to keep. Our general approach (detailed below) was to maximise both dimensionalities separately, in order to obtain the richest possible description of multiple population modes. However, this needs to be done with the constraint that reported modes are statistically robust (i.e. avoiding over-fitting).

Therefore, starting from 3913 columns (IDPs) in $W$, we ran PCA and ICA at dimensionalities from 60 to 150, evaluating each with respect to a metric of split-half reproducibility (all code for this is available, as described above). For each PCA dimensionality reduction, this test of reproducibility applies the following procedure: ICA is run three times - first with all data, and then twice on randomly-split-halves of the data; the components from the two split-half runs are then ordered according to best-match (via the Hungarian greedy-pairing algorithm) to the all-data ICA run; correlation between the split-half paired ICA components' source (IDP) vectors was estimated, and only extremely similar components ($r>0.9$, see below for estimation of the associated statistical significance) were retained; all the above steps were run 10 times (each with a different split-half-subjects

randomisation) and averaged together to give the reproducibility test-statistic - the number of reproducible components estimable by the current dimensionality.

The PCA dimensionality resulting in the largest number of highly reproducible components was found to be 128, and from this, 62 ICA components were highly reproducible. Finally, ICA was rerun with this PCA dimensionality 30 times, each time with random split-half-subjects, and the most robust run (in terms of reproducibility) was then utilised, resulting in the final set of 62 ICA components.

As a simple highly conservative test of significance, we computed null correlations between an 'IDP-weight vector' of random noise of 62 samples (the minimum possible degrees-of-freedom, and hence the most conservative test) and 128 other random vectors, taking the maximum correlation magnitude across all 128, and then building up the null distribution of this maximum across 1 million random null tests. The maximum across all 1 million only reached $|r| = 0.68$ (90th percentile $|r| = 0.41$), whereas we are only keeping modes with split-half reproducibility $|r| > 0.9$. We can therefore be confident that the final components are robustly present with a significance of at least $P < 10^{-6}$ (and probably much greater).

As a second test of significance of the overall data-driven modelling, including the age-weighting of inputs to the PCA+ICA, we applied the following null evaluation. We used a random vector instead of age to carry out the IDP weighting, ran ICA at dimensionality of 128, and correlated all 128 resulting subject-weight-vectors with the random vector, recording the maximum correlation magnitude across all 128 (and by doing so making this more conservative than by testing just 62 modes chosen through split-half reproducibility). From 100 random repeats of this test, the maximum absolute correlation (across 100 repeats and 128 ICA modes) was just $|r| = 0.032$ (to be compared against the age correlations shown in *Figure 1—figure supplement 2D*).

A given ICA component is unchanged in its modelling of the input data if the sign of both the subject-weights-vector and the IDP-weights-vector are inverted (as these two inversions cancel each other out - the initial sign of each is arbitrary, as with PCA). Hence we oriented the 62 modes of population variability so that their subject-weights-vectors were all positively correlated with age, in order for simplicity of later interpretation.

We next investigated whether the 62 modes of brain aging could be arranged in fairly clean clusters having similar patterns of aging; if so, this could aid in simplifying interpretation of the modes. *Figure 1—figure supplement 1* shows hierarchical clustering of the correlation matrix of subject-weight-vectors. The reasonably strong diagonal-block-structure suggests that a lower-dimensional clustering could be a useful way to help simplify the interpretation of the 62 modes of brain aging. Therefore, in order to carry out a lower-dimensional analysis, we re-ran the ICA, this time on PCA dimensionalities running from 2 to 50 (from the same IDPs matrix that was fed into the higher dimensional mode estimation above). We evaluated objectively which dimensionality provided the cleanest clustering of the 62 modes, by optimising the following cost function: We estimated the correlation matrix of 62 modes' subject-weight vectors with each low-dimensional ICA set of subject-weight vectors, took the magnitude of this, sorted each column (spanning the low-dimensional analysis), subtracted the second-strongest correlation from the first, and summed this over columns (the high-dimensional components). This cost function therefore describes how cleanly each high-dimensional mode is associated with just a single low-dimensional component. We found that the optimal lower dimensionality was 6.

As well as being sign-oriented to positively correlate with age, the modes (from both 62 and 6 dimensionalities) were ordered (numbered) according to decreasing correlation with age, again for convenience of interpretability and with no loss of generality in the modelling. We refer to the higher-dimensional modes of aging via their (ordered) number with subscript 62 (e.g. 'brain aging mode $2_{62}$'), and lower-dimensional mode-clusters via their number with subscript 6 (e.g. 'brain aging mode-cluster $3_6$'). *Figure 1—figure supplement 1B* shows the correlation matrix between subject-weight-vectors from the two dimensionalities, with the fairly clear clustering visible (i.e. most of the 62 modes are strongly associated with at most one of the 6 mode-clusters).

## Brain-age visualisation and sex-separated aging curves

We used the estimated population modes to model brain aging, following the general regression-based approaches laid out in *Smith et al. (2019)*.

For simple visualisation of each mode's overall age dependence, we utilise the simple 'switched' model, where imaging measures are characterised as a function of age. We used an age model with linear, quadratic and cubic powers of age, to fit to each mode's subject-weights-vector. The fitted age curves for all 62 modes are shown in *Figure 1*, as well as the raw data (scatterplot points, one per subject) and fitted curves for two example modes. By definition (see above), all modes have positive age correlation, although for some modes these positive coefficients are close to zero. *Figure 1—figure supplement 1C* shows the equivalent fitted age curves for the 6 mode-clusters. *Figure 1—figure supplement 2A* shows the ratio of the standard deviation explained by the mean-age-dependent-curves to the standard deviation of the data (the mode subject-weights). There is a continuous distribution of ratio values, from above 0.6 in the lowest-numbered modes, through to almost zero for the highest-numbered modes (though all of the mode-clusters are above 0.3). (Significance testing on strength of age dependence is reported below).

*Figure 1—figure supplements 3–9* show, for each mode, sex-separated aging curves, and also the aging curves for the unique variance captured by each mode. For the latter, the subject-weights-vector for each mode is 'partialled' - that is, has all other modes' subject-weights-vectors regressed out, before re-fitting the average age curves for visualisation in the figures. For these sex-separated aging curves, sex-separated subject-weight-vectors were first estimated, by multiplying the ICA IDP weights matrix into a version of the original data matrix that had all confounds removed as before, but this time without including sex as one of the confounds. Averaged age-curves were then generated; for these visualisations, sex-separated age curve fitting was carried out in a more model-free way than the parametric (cubic) age model used for our more quantitative analyses. Specifically, for the purposes of showing the data in a more raw form, we simply use sliding windows of width 5y to average (sex-separated) data points around each 1y age bin centre (although averages of the two sex-separated curves are visibly highly consistent with the cubic average age model shown underneath in grey). For the majority of modes, the two sexes have highly similar age curves, but for some (e.g. mode $1_{62}$), there are strong differences.

As one would expect, the age dependence is less strong in the partialled modes, as each has a large amount of shared variance regressed out. Some even show negative overall age dependence after partialling (e.g. mode $22_{62}$).

## Brain-age delta modelling

For our quantitative modelling of brain-age delta (estimated brain-age minus actual age), we use the common approach of modelling age as a function of imaging features (as opposed to the other way round as above), combined with the second step from *Smith et al. (2019)*, which removes age-related bias in the brain-age delta. Hence, for the first step, one would model

$$Y = X\beta_1 - \delta_1, \tag{1}$$

where $Y$ is age, $X$ is the modes' subject-weights matrix (size $N_{subjects} \times N_{modes}$), $\beta_1$ is the ($N_{modes} \times 1$) vector of regression parameters, and $\delta_1$ the initial estimate of brain-age delta. The above produces a $\delta$ that is orthogonal to $X$ (the imaging measures) rather than $Y$ (age). Thus, we can think of the first stage residuals, $\delta_1$, as the aspects of age that cannot be accounted for by the imaging measures. The second stage of modelling aims to refine this model by identifying aspects of this first-stage $\delta_1$ that cannot be accounted for by age terms or confounds. Note that this stage explicitly forces $\delta_2$ to be orthogonal to all of the components in $Y_2$, including age:

$$\delta_1 = Y_2\beta_2 + \delta_2, \tag{2}$$

where the regression matrix $Y_2$ includes not just linear, quadratic and cubic age terms, but also the other confound variables. One can equivalently view the first step above as a sum over modes:

$$Y = \sum_i (X_i\beta_{1i} - \delta_{1i}), \tag{3}$$

where we have separated out the contributions to the modelling from each mode, along with breaking down the delta into a delta vector per mode. The $\beta$ regression parameters remain determined by the standard multiple regression inversion, $\beta_1 = (X'X)^{-1}X'Y$, and each $\delta_{1i}$ is estimated simply as

$X_i\beta_{1i} - aY$. Here $a$ is an arbitrary scaling (e.g. $1/N_{modes}$) whose value is not important because the term $aY$ will be removed by the second step that regresses out age and confounds. One can then keep the second step deltas also separated:

$$\delta_{1i} = Y_2\beta_{2i} + \delta_{2i}, \tag{4}$$

The original $\delta_1$ is the sum of the individual modes' $\delta_{1i}$ vectors, and $\delta_2$ is the sum of all modes' $\delta_{2i}$ vectors. By separating out each mode's contribution to the overall brain aging delta, and by doing so in the context of the modelling being an 'all-in-one' multivariate model (multiple regression using all modes' subject-weights vectors), we are able to then go on to study how the different modes' brain-aging are distinct from each other, as well as how they combine to give an overall best-estimate of brain age. The combined modelling across all modes (summed $\delta_{2i}$) results in a mean absolute 'error' of 2.9y.

As with the partialled subject-weight-vectors described above, we also generate partialled versions of the modes' deltas; for each mode's $\delta_{2i}$, we regress out all of the others. We can then, for example, correlate these partialled delta estimates with non-imaging variables in order to find associations with the unique variance in each mode's brain-aging delta.

In *Figure 1—figure supplement 2B,C*, we show the standard deviation (variation across subjects) associated with the individual modes' brain-age modelling from step 1 ($X_i\beta_{1i}$), the deltas after step 2, and the partialled deltas. There is not a qualitative difference between the three curves, because the $\beta$ regression parameters are driven by the unique variance components of the original modes' subject-weight-curves. There is not (expected to be) a simple relationship between the original strength of age dependence for a given mode, and the age dependence in its unique variance; this also explains why the curves are not monotonically decreasing (as they clearly are, by definition, in the univariate analyses shown in *Figure 1—figure supplement 2A*).

In *Figure 1—figure supplement 2D,E*, we show related information - statistics from the multiple regression in the age modelling first step (as well as the simple univariate correlation between individual modes' subject-weight-vectors and age, for reference). The regression $\beta$ values vary highly from mode to mode (as mentioned above), driven by the unique variance in each mode. Several modes have negative $\beta$ weights, meaning that their unique variance is negatively associated with age, even though their original correlation with age was (by definition) positive. Two modes ($22_{62}$ and $50_{62}$) have quite strongly negative $\beta$ (more negative than $-0.5$).

## Non-additive brain-age delta estimation

Following the approach outlined in *Smith et al. (2019)*, we estimated the extent to which the scale (size) of delta changes across the age range present in the UK Biobank data. This is a distinct model from those outlined above, which treat delta as additive to age (to form brain age), and hence being constant in overall scale (as a function of aging). This would represent not a simple shift in brain age, but potentially (e.g.) something like an acceleration in aging (delta gets bigger with age). Of course, with a limited range of ages, such a scaling term might be effectively captured with a purely additive term, so this modelling is really asking whether our data show evidence for a scaling effect, rather than making a strong statement about the form deltas take over the entire age range.

The results are shown in *Figure 1—figure supplement 10*. 17 modes and three mode-clusters show statistically significant amount of non-additive brain aging. In most cases, delta is increasing with age (e.g. as can be seen visually in *Figure 1—figure supplement 10C* for mode $4_{62}$, but some modes are decreasing (e.g. as seen in *Figure 1—figure supplement 10D* for mode $11_{62}$).

## Brain-age modes' structural and functional interpretation

The raw ICA IDP-weights-vectors are plotted in *Figure 2—figure supplement 1*, with IDPs running along the x axis. FreeSurfer-derived structural IDPs are to the right, functional connectivity (from resting-state fMRI) estimates in the central portion (this is largely - but not completely - empty), and other structural, diffusion MRI and task fMRI measures in the left-most block. These are the raw weights, and we do not discuss this visualisation in greater detail here, because the more compact summary of IDP weights in *Figure 2* is more interpretable, and also the full lists of strongest weights are provided in spreadsheets (see Data availability).

*Figure 2* arranges IDPs into logical groupings of distinct types of measures ('modality types' - for the full list of IDPs and their modality groups, see Data availability). For each modality group $j$ comprising $N_j$ IDPs, the $N_{modes} \times N_{IDPsj}$ matrix is fed into ICA to reduce the number of IDPs to a more visually-compact form of IDP 'clusters' - thus each column in the figure represents a group of IDPs with similar behaviour across modes. For each IDP modality group, the number of displayed components is data-dependent, utilising the PCA eigenspectrum to determine ICA dimensionality and then retaining ICA components with sufficiently strong maximum weight, though always displaying at least one strongest component (see code linked in Data availability for full implementational details).

We show separate visualisations for the 62 modes and also the 6 mode-clusters, with the same IDP groupings for each (but separate ICA decompositions, as we did not want either decomposition to influence the other). We can see many clear correspondences between the modes and mode-clusters in compatible ways to those described above. For example, modes $5_{62}$ and $10_{62}$ and mode-cluster $3_6$ relate closely to each other, and all are driven by T1 contrast across the grey-white boundary. These figures are discussed in greater detail in Results.

Finally, voxelwise mapping of deltas were estimated to help interpret some modes and relevant imaging modalities. In some cases, it was found to be useful to simply correlate delta against the $N_{subjects} \times N_{voxels}$ full imaging data, and in other cases we averaged the images from the 1000 subjects having the lowest (e.g. most negative) delta values, and separately averaged the 1000 subjects with the largest values, to generate two average images for direct visual comparison. Where appropriate, the imaging data was deconfounded (across-subjects) using the same confound regressors as described above.

## Associations of brain-age Delta with non-imaging measures

We utilised 8787 non-imaging, non-genetic measures (which we refer to here as nIDPs - non-imaging-derived phenotypes) from UK Biobank, spanning 16 groups of variable types, including early life factors (such as being breastfed as a baby), lifestyle factors (e.g. exercise, food, alcohol and tobacco variables), physical body measures (e.g. body size, fat, bone density variables and blood assays), cognitive test scores, and health (including mental health) variables (see *Figure 3—figure supplements 2–3* and online spreadsheets described in Data availability). These variables were automatically curated using the freely available FUNPACK (the FMRIB UKBiobank Normalisation, Parsing And Cleaning Kit https://git.fmrib.ox.ac.uk/fsl/funpack) software; this sorts variables into hand-curated groups, ensures that quantitative variable codings are parsed into at least monotonically-sensible values, and separates categorical variables into multiple binary indicator variables.

The nIDPs were then passed through similar preprocessing as above for IDPs; they were quantile normalised and had all confounds regressed out (including age-related confounds). The one difference here was that, to avoid statistical instability when working with variables that only exist for one sex (e.g. related to menopause), the confound variables were sex-separated before being applied.

The UK Biobank nIDPs have varying amounts of missing data. Here, we used 8787 variables having data from 40 subjects or more. Therefore, the full set of associations of nIDPs against brain-age delta have widely-varying degrees-of-freedom, and taking into account correlation p-values is important (and not just correlation $r$ values). The histogram of non-missing data proportions is shown in *Figure 2—figure supplement 2*.

To identify the strongest associations between brain-age delta (for each mode and mode-cluster), we used simple Pearson correlation (as described above, both IDPs and nIDPs have been quantile-normalised, that is, Gaussianised). For each mode/mode-cluster, we computed correlations between nIDPS and the delta estimates, and also partialled delta estimates (to identify associations between nIDPs and the unique variance in the deltas). We also computed the same sets of associations for just females and just males. In detailed spreadsheets (see Data availability), we report all associations where any of the tests (i.e. using all subjects, and just females, and just males) have a significance value of $-\mathrm{Log}_{10}P > 5$, although these should be interpreted in the light of the fact that conservative Bonferroni correction across 62 modes and all nIDPs would have a $-\mathrm{Log}_{10}P$ threshold of 7.0, while across 6 mode-clusters this would be 6.0.

Summary plots simplifying the mapping of modes onto nIDP variables and variable groups (using variable-group-specific ICA) were created in the same manner as described above for IDPs, and form part of *Figure 3—figure supplements 2–3*.

## GWAS of brain-age delta

We carried out genome-wide association studies (univariate regressions) of all delta estimates, following the approach used in *Elliott (2018)*. We used the second UK Biobank release of imputed genetic data, comprising over 90 million structural variants (which are primarily SNPs, and are referred to here in general as SNPs for brevity).

We used a minor allele frequency (MAF) threshold of 1%, imputation information score threshold 0.3 and Hardy-Weinberg equilibrium P-value threshold $10^{-7}$. We reduced the subjects used for GWAS to a maximal subset of unrelated subjects with recent British ancestry (to avoid the confounding effects of gross population structure and complex cross-subject covariance). Relatedness was determined by thresholding the kinship matrix at 0.175, and recent British ancestry was determined using the variable in.white.British.ancestry in the provided genetic data files. 40 population principal components (as supplied by UK Biobank) were used as GWAS confound regressors (again, to avoid the confounding effects of gross population structure).

This QC filtering resulted in a total of 9,812,242 SNPs and 15,952 subjects (samples), which we partitioned at random into a 10,612 subject discovery sample and a 5,340 subject replication sample. GWAS was carried out using BGENIE v1.2 (https://jmarchini.org/bgenie/).

The standard single-phenotype GWAS threshold is $-\mathrm{Log}_{10}P = 7.5$. Our Manhattan plots (of significance vs. SNPs) show this threshold as well as an adjustment of this for the Bonferroni factor of 62 +6 phenotypes, i.e., $-\mathrm{Log}_{10}P = 9.33$. This is likely conservative due to correlations across phenotypes (modes and mode-clusters).

After performing the GWAS, we used a method described in *Elliott (2018)* to identify meaningfully distinct lead (peak) SNPs, taking into account correlation amongst neighbouring SNPs (linkage disequilibrium). In effect, this identifies distinct clusters of significantly associated SNPs. This method works by forming a set containing all of the significant SNPs, and then iteratively retains only the top-most significant hit among all SNPs in the set while removing other SNPs within 0.25 cM (approximately 250kbp on average) of the reported peak SNP, terminating after all significant SNPs are removed or retained for reporting.

*Figure 3* shows various Manhattan plots for individual delta estimates as well as all-in-one estimates. Individual Manhattan plots for every mode/mode-cluster, both sex-combined and sex-separated, and for delta and partialled delta, were generated (see Data availability). Summary plots simplifying the mapping of modes onto SNPs and chromosomes (using variable-group-specific ICA) were created in the same manner as described above for IDPs and nIDPs, and form part of *Figure 3—figure supplements 2–3*.

Finally, we ran several distinct kinds of additional genetic analyses. Using the Genome Browser https://genome.ucsc.edu we manually identified RSIDs for all indel variants that we found to have peak associations (that is, all peaks for all modes and all mode-clusters). We then used FUMA https://fuma.ctglab.nl (*Watanabe et al., 2017*) to map SNPs/variants to genes. Next, we used FUMA in conjunction with ANNOVAR http://annovar.openbioinformatics.org/en/latest/ (*Wang et al., 2010*) to identify SNPs in LD with the peak SNPs, and to functionally annotate them. Taking advantage of gene expression and chromatin databases, we identified eQTL and chromatin mappings/interactions for SNPs and genomic loci, again via FUMA. We also carried out PHEWAS, identifying details of other traits from previous (largely non-UK Biobank) studies having associations with our peak SNPs, via the PHEWAS-atlas/FUMA tool. Finally, we used LD score regression with LDSC v1.0.1 (*Bulik-Sullivan et al., 2015*) (applied separately to each mode/mode-cluster) to estimate genetic (SNP) heritability of all modes and mode-clusters, as well as to estimate genetic co-heritability with Alzheimer's disease (*Lambert et al., 2013*) and Parkinson's disease (*Simón-Sánchez et al., 2009*) (see online supplemental materials for AD/PD summary statistics information detail and acknowledgements). All LDSC analysis was done with LD scores computed using the 1000 Genomes European (EUR) subjects (*Auton et al., 2015*).

## Acknowledgements

SS is supported by a Wellcome Trust Strategic Award 098369/Z/12/Z and a Wellcome Trust Collaborative Award 215573/Z/19/Z. KM is supported by a Wellcome Trust Senior Research Fellowship 202788/Z/16/Z. The Wellcome Centre for Integrative Neuroimaging (WIN FMRIB) is supported by centre funding from the Wellcome Trust (203139/Z/16/Z). GD is supported by an MRC Career

Development Fellowship (MR/K006673/1). This research has been conducted in part using the UK Biobank Resource under Application Number 8107. We are grateful to UK Biobank for making the data available, and to all UK Biobank study participants, who generously donated their time to make this resource possible. Analysis was carried out on the clusters at the Oxford Biomedical Research Computing (BMRC) facility and FMRIB (part of the Wellcome Centre for Integrative Neuroimaging). BMRC is a joint development between the Wellcome Centre for Human Genetics and the Big Data Institute, supported by Health Data Research UK and the NIHR Oxford Biomedical Research Centre.

## Additional information

### Funding

| Funder | Grant reference number | Author |
|---|---|---|
| Wellcome | 203139/Z/16/Z | Stephen M Smith<br>Karla L Miller |
| Wellcome | 098369/Z/12/Z | Stephen M Smith |
| Wellcome | 215573/Z/19/Z | Stephen M Smith |
| Wellcome | 202788/Z/16/Z | Karla L Miller |
| Medical Research Council | MR/K006673/1 | Gwenaëlle Douaud |

The funders had no role in study design, data collection and interpretation, or the decision to submit the work for publication.

### Author contributions

Stephen M Smith, Conceptualization, Data curation, Formal analysis, Supervision, Funding acquisition, Validation, Investigation, Methodology, Project administration; Lloyd T Elliott, Conceptualization, Data curation, Software, Formal analysis, Validation, Investigation, Visualization, Methodology, Project administration; Fidel Alfaro-Almagro, Data curation, Software, Formal analysis, Validation, Investigation, Visualization, Methodology; Paul McCarthy, Thomas E Nichols, Data curation, Software, Formal analysis, Validation, Investigation, Methodology; Gwenaëlle Douaud, Conceptualization, Data curation, Formal analysis, Validation, Investigation, Visualization, Methodology; Karla L Miller, Conceptualization, Resources, Data curation, Software, Formal analysis, Supervision, Funding acquisition, Validation, Investigation, Visualization, Methodology, Project administration

### Author ORCIDs

Stephen M Smith (iD) https://orcid.org/0000-0001-8166-069X
Gwenaëlle Douaud (iD) http://orcid.org/0000-0003-1981-391X
Karla L Miller (iD) http://orcid.org/0000-0002-2511-3189

### Ethics

Human subjects: The UK Biobank has approval from the North West Multi-centre Research Ethics Committee (MREC) to obtain and disseminate data and samples from the participants (http://www.ukbiobank.ac.uk/ethics/), and these ethical regulations cover the work in this study. Written informed consent was obtained from all participants.

### Decision letter and Author response

Decision letter https://doi.org/10.7554/eLife.52677.sa1
Author response https://doi.org/10.7554/eLife.52677.sa2

## Additional files

### Supplementary files

• Transparent reporting form

## Data availability

All subject-level data (IDPs, nIDPs and genetics) are available upon application to UK Biobank. The UK Biobank data acquisition MRI protocol, and the image processing and IDP generation pipelines are all freely available (https://www.fmrib.ox.ac.uk/ukbiobank). Additional resources relating to group-average image analysis can be found at https://www.fmrib.ox.ac.uk/ukbiobank/. This includes population-average templates for all of the different imaging modalities, and lists/images of all rfMRI nodes and edges. All code developed for the work reported here (Matlab) is freely available from https://www.fmrib.ox.ac.uk/ukbiobank/BrainAgingModes. The same website also contains the following additional supplemental materials: Figures with all modes'/mode-clusters' individual GWAS Manhattan plots; GWAS summary statistics files; rfMRI summary brain images showing visually the brain regions ('nodes') and pairs of brain regions ('edges') significantly associated with all modes and mode-clusters; tables/spreadsheets listing all IDPs used, the strongest nIDP associations with all modes/mode-clusters, the strongest IDP weights for all modes/mode-clusters, and the peak GWAS associations (tables can be downloaded or viewed online); and additional genetic analyses including functional annotation, gene expression, associated traits from previous GWAS studies, and genetic heritability/co-heritability results.

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

## Appendix 1

### Supplementary comments on body size and other 'baseline' causal factors in IDPs and brain aging

We now include a simple discussion of the opposing signs of involvement of the various body-size-related variables seen for mode-clusters $1_6$ and $2_6$.

The typical starting point for modelling brain aging (e.g., see **Smith et al., 2019**) is

$$Y_B = Y + \delta = f(X) = X\beta, \tag{5}$$

where actual age is $Y$ (an $N_{subjects} \times 1$ vector), brain age is $Y_B$ and the brain-age delta is $\delta = Y_B - Y$. The imaging data matrix is $X$, which has $N_{subjects}$ rows and $D$ columns; the columns are features from the imaging data, and might be different voxels, or different IDPs (imaging-derived phenotypes - summary measures of brain structure and function), or different modes.

Here we treat $X$ as a single feature, for example, total volume of grey matter. We might expect grey matter volume $G$ for subject $i$ to depend both on overall body size as well as age-related atrophy, and hence follow a form like:

$$G_i = bB_i - Y_i(\alpha_{average} + \alpha_i) \tag{6}$$

where $B_i$ is a subject's 'baseline' body size, $b$ the coefficient relating body size to grey matter volume, $\alpha_{average}$ is the population average rate of atrophy (the reciprocal of $\beta$ in general), and $\alpha_i$ is the subject's deviation (in atrophy rate) from the population average. By definition here $b$ and $\alpha_{average}$ are positive.

Now, in such cases where the imaging feature is negatively correlated with age (hence the minus sign above), the mode preprocessing used in our modelling flips the sign of the mode so that the subject weights are positively correlated with age (see Materials and methods). Hence we have:

$$X_i = -G_i = -bB_i + Y_i(\alpha_{average} + \alpha_i) \tag{7}$$

$$X_i\beta = -b\beta B_i + Y_i + Y_i\alpha_i\beta \tag{8}$$

$$\delta_i = X_i\beta - Y_i = \Delta_i - b_2 B_i, \tag{9}$$

where $b_2 = b\beta$ (i.e., is typically a positive coefficient, although multiple-regression age prediction from multiple modes can result in negative $\beta$, as discussed above) and $\Delta_i = Y_i\alpha_i\beta$ is the aspect of the brain age delta that is separate from the effect of the baseline body size (i.e., relates to the ongoing atrophy).

Hence estimated $\delta$ does correctly reflect the atrophy-related *delta*; however, additionally, between-subject variations in baseline body size result in a larger body giving an apparently lower $\delta$. In cases where the IDP/mode changes are positively correlated with aging (e.g., CSF volume, as in mode-cluster $1_6$), there is no negative sign above, and no sign-flipping for the mode, and hence the apparent effect of body size is not reversed. Of course, to further complicate matters, some 'baseline' or 'background' factors (such as socio-economic status) may well have a significant causal role both in baseline IDP/mode values as well as aging rate.

Put more simply and qualitatively, a subject with large body size will have large baseline CSF, and the brain-age modelling will therefore likely consider that large body size is a 'bad thing' with respect to mode-cluster $1_6$; on the other hand, the same subject will have large baseline grey matter, and the brain-age modelling will therefore consider that large body size is a 'good thing' with respect to mode-cluster $2_6$. For such cases of course neither simplistic conclusion is appropriate.

Note that in the simpler case where an nIDP is more directly related to an IDP or mode (for example, as is found with alcohol and smoking), the signs of the associations between $\delta$

and the IDP and the nIDPs are all simply consistent and easily interpretable. For example, for mode-cluster $1_6$, CSF volume is positively correlated with $\delta$ (higher CSF volume is indeed a 'bad thing'); for mode-cluster $2_6$, grey matter volume is negatively correlated with $\delta$ (grey matter volume is a 'good thing'), and for both mode-clusters, alcohol and smoking are positively correlated with $\delta$ (they are both 'bad things').

