## [Decision Letter]

**Acceptance summary:**

This paper uses a large community based sample of brain imaging, genetic, and lifestyle data to identify different trajectories of aging in brain structure and function. In addition to identifying relationships across specific brain networks, the work also emphasizes the importance of viewing brain aging as a combination of interactive influences, as opposed to a unitary process.

**Decision letter after peer review:**

Thank you for submitting your article "Brain aging comprises many modes of structural and functional change with distinct genetic and biophysical associations" for consideration by *eLife*. Your article has been reviewed by three peer reviewers, and the evaluation has been overseen by a Reviewing Editor and Floris de Lange as the Senior Editor. The following individuals involved in review of your submission have agreed to reveal their identity: Christopher Madan (Reviewer #2); Lars Nyberg (Reviewer #3).

The reviewers have discussed the reviews with one another and the Reviewing Editor has drafted this decision to help you prepare a revised submission.

Summary:

The authors take a novel approach to brain aging by looking at associations across neuroimaging markers to identify patterns that covary with age in a large dataset. These brain "modes" can then be used to predict chronological age. Importantly, separating brain markers into dissociable sets the stage for identifying complementary physiological processes that occur with aging and how these processes relate to genetic, environmental, or outcome measures.

1) Although the constraints of large-scale data collection are understandable, concerns about the reliability of functional MRI IDPs should be addressed: task-related and resting state fMRI with <20 minutes of data post data cleaning frequently produce relatively unreliable metrics (i.e., task: https://www.biorxiv.org/content/10.1101/681700v1: PMID: 28757305), that are not sufficiently reliable for individual differences research. Please provide rationale for the imaging derived phenotypes included and in particular, why phenotypes with such low reliability are included. It would also be useful to generate components using only imaging derived phenotypes that have been shown to be sufficiently reliable for individual differences research. Minimally, data suggesting that task-related and resting state fMRI (at the time collected) have poor reliability should be discussed.

2) The current findings depend on the specific IDPs generated by the authors. To what degree would the IDPs (and thus identified modes, and the following conclusions) be replicable by other analysis teams? Similarly, would the identified IDPs and modes be present in a different dataset (or with different imaging modalities)? Although any additional analyses would be welcome, at a minimum, these considerations should be discussed clearly with an eye towards future reproducibility and interpretation.

3) The conceptual framing and implications need adjustment to be more nuanced. The identification of modes is a useful perspective, but also raises some important questions. Can modes age at different rates within in an individual? How can modes/clusters be integrated within an individual? The multivariate nature of a mode-based view of brain aging seems to be difficult to interpret as "brain age" given the high dimensional space and cross-sectional nature of the data (related to point 2 above). Some additional comment on these considerations would be welcome, again, both for the current study and also for guiding future discussions of these important issues.

The genetic analyses are reasonable, but could be considerably strengthened. The below suggestions are some examples – it is not the case that all of the suggested additional analyses are required for publication, but some additional work to strengthen the genetic component of the work (which is, obviously, very important) will be needed.

4) First, it would be useful to further replicate associations with expression using other expression datasets (e.g., common mind, depression genes and network study) and note whether any are brain region (i.e., tissue type) specific. Further, testing whether genomic risk for these brain aging phenotypes is correlated with gene expression using a technique such as TWAS or predixcan would further add informativeness and novelty to the GWAS-aspect of this paper (see: https://bogdan.dgsom.ucla.edu/pages/twas/ and/or: https://github.com/hakyimlab/PrediXcan). Second, additional follow-up annotation of GWAS results would greatly improve this manuscript. In particular, FUMA (https://fuma.ctglab.nl/) is an invaluable and easy to use resource. It would be useful to conduct more comprehensive analyses (e.g., gene-based testing) and annotation (chromain, more distal eQTLsof GWAS findings) of GWAS results.

5) You currently note several phenotypes that identified SNPs are correlated with. Much of this from the UK biobank – while this is interesting, it would be useful to explore associations in other datasets as relying on UK biobank results could be somewhat circular. An online tool – GWAS atlas PheWAS has been developed for this purpose: https://atlas.ctglab.nl/PheWAS.

6) It is now widely accepted that complex behavioral phenotypes, including neural ones (e.g., PMID: 26854805) are incredibly polygenic – consistent with such data, it would be useful to consider moving beyond probing single variant associations and more comprehensively examine whether the polygenic architecture of these brain age phenotypes is correlated with other phenotypes within the literature (e.g., longevity, Alzheimer's, obesity, etc.) using LD score regression (https://github.com/bulik/ldsc). The LD hub online tool may be useful for such analyses: http://ldsc.broadinstitute.org/. It would also be useful to report SNP-based heritability for these different phenotypes.

---

## [Author Response]

[…] 1) Although the constraints of large-scale data collection are understandable, concerns about the reliability of functional MRI IDPs should be addressed: task-related and resting state fMRI with <20 minutes of data post data cleaning frequently produce relatively unreliable metrics (i.e., task: https://www.biorxiv.org/content/10.1101/681700v1: PMID: 28757305), that are not sufficiently reliable for individual differences research. Please provide rationale for the imaging derived phenotypes included and in particular, why phenotypes with such low reliability are included. It would also be useful to generate components using only imaging derived phenotypes that have been shown to be sufficiently reliable for individual differences research. Minimally, data suggesting that task-related and resting state fMRI (at the time collected) have poor reliability should be discussed.

We have now carried out this analysis (see below), and expanded on relevant discussion points.

The reliability of IDPs depends on many factors, including the modality, the exact preprocessing and processing used, the quality/quantity of data, the potential confounding effects (separately from raw SNR), and the goal at hand; while we agree with the reviewer’s premise that in some studies some IDPs have been shown to be relatively noisy (or unreliable), we do not believe that a “binary” judgement of any given class of IDPs is necessarily appropriate. On the one hand, we agree that the UK Biobank resting-fMRI connectivity edges and task-fMRI activation IDPs are overall more noisy than most other classes of IDPs in UKB. But, on the other hand, there is strong evidence that they contain meaningful information in the context of group-level analyses, e.g., as reflected in significant (and cross-validated) associations with cognitive test scores (Miller et al., 2016). In another example, while individual fMRI “edge” IDPs showed weak heritability and genetic association, some latent factors derived from the same IDPs (6 ICA components derived just from rfMRI edges and no other information) showed strong and significant (and replicated) heritability and genetic associations (Elliott et al., 2018). As a third example, we have shown that, out of the full set of functional and structural IDPs in UKB, only these resting-fMRI edges showed significant correlation with handedness (Wiberg, Brain, 2019).

Moreover, if valid statistical testing produces significant results for a given class of IDP, then by definition those significant results are valid, and relatively unreliability (“noisiness”) compared to a different class of IDP is simply that: *relative* unreliability, and not on its own a sound reason to discount or exclude those results. Note that the above-mentioned publications describe the exact fMRI IDPs used here (and their rationale), and demonstrate that they do contain usable meaningful signal.

There is of course the important distinction between the level of reliability at the individual level vs the population level, where working with the latter can naturally result in a large boost to the effective SNR of any given calculation using a class of IDPs. The brain-aging modes derived here are estimated from thousands of subjects’ data, greatly enhancing reliability for these patterns found overall (which is supported by the various strict reproducibility results described in our Materials and methods. While it might be desirable to be able to make strong inference at the individual level (for pretty much any imaging study, but certainly in clinical practice), this is not a pre-requisite for the vast majority of studies, which make statements at the group level. We do agree that **if** it is required for a given measure to be reliable for an individual subject, then ICC would need to be high, and we also agree that the field of neuroimaging may currently assume that the reliability of many fMRI measures is higher than it actually is. We also agree with the reviewers that it is useful to add further discussion of the relative “reliability” of the different classes of IDPs present in our paper. We have now added this to the central part of the Discussion (sixth paragraph).

As requested, we have rerun the brain-mode-clusters ICA without including any fMRI IDPs. We then correlated the IDP-weight ICA vectors against those original vectors that were not originally dominated by fMRI IDPs (i.e., excluding cluster 5), after greedy pairing. These vectors were virtually unchanged (all paired correlations being r>0.9). We include these results in the above-mentioned Discussion section.

2) The current findings depend on the specific IDPs generated by the authors. To what degree would the IDPs (and thus identified modes, and the following conclusions) be replicable by other analysis teams? Similarly, would the identified IDPs and modes be present in a different dataset (or with different imaging modalities)? Although any additional analyses would be welcome, at a minimum, these considerations should be discussed clearly with an eye towards future reproducibility and interpretation.

The paper’s Introduction includes the sentences: *“For this work we used 3,913 IDPs (imaging-derived phenotypes, generated by our team on behalf of UK Biobank, and made available to all researchers by UK Biobank)” and “All data is available upon application to UK Biobank”.* As with our previous papers (Miller, 2016, Elliott, 2018), we have used IDPs that are already available to the scientific community (indeed, we are not allowed to use these in “our own” research until they have been made publicly available).

Additionally, the entire processing pipeline (that carries out image pre-processing and derives the IDPs) is freely available on Github, as is the full data acquisition MRI protocol (apologies for not noting this in the paper previously; we have now added this information to the Materials and methods and to Subsection “Data and code availability, and additional supplementary tables and figures”). The paper already states that the *new* code developed for this paper is publicly available.

Hence, all input data to this work is openly available, and all results from this work are certainly replicable by other analysis teams. We hope these statements in the manuscript will make this sufficiently clear to readers.

Similarly, for data from other (non-UKB) studies, the full code is available for deriving the exact same set of IDPs, as long as the same imaging modalities are acquired. How well harmonised those IDPs would be with UKB IDPs would of course be a “sliding-scale”, dependant on how similar the MRI scanner hardware, scanner software and protocol were to those in UKB. Similarly, how similar derived brain modes would be to those that we report here would likewise be a sliding-scale, dependant on how well-matched the data characteristics (and subject group demographics) were.

We have now added further discussion of these points near the start of Discussion.

3) The conceptual framing and implications need adjustment to be more nuanced. The identification of modes is a useful perspective, but also raises some important questions. Can modes age at different rates within in an individual? How can modes/clusters be integrated within an individual? The multivariate nature of a mode-based view of brain aging seems to be difficult to interpret as "brain age" given the high dimensional space and cross-sectional nature of the data (related to point 2 above). Some additional comment on these considerations would be welcome, again, both for the current study and also for guiding future discussions of these important issues.

In the models we use here, brain-aging modes can indeed, in theory, age at different rates in an individual:

if “aging rate” is reflected in the delta, then every mode in every individual has a separately estimated value. The reviewers touch on an important point: the distinction between different modes (in different subjects) having a **fixed** constant delta relative to the population norm curve, vs. an accelerated or decelerated aging **rate**. This is mentioned towards the end of Discussion.

In the context of brain-age modelling, multiple modes can be integrated within an individual, for example through the all-in-one modelling presented in the paper. This uses all modes to form a single predictor of age, as presented (e.g.) in the various GWAS figures. While this is one straightforward answer to the question of integration, we agree that this is an important question, given that results like the GWAS plots suggest that our single predicted age estimate has over-reduced the modelling, “diluting” out the rich set of associations found with single modes. On the other hand, careful consideration of the all-in-one model parameters (regression betas) is complex and potentially biologically informative (see Figure 1—figure supplement 2 and all sex-separated partialled mode aging curves, as well as Discussion, penultimate para).

We have now expanded on and clarified these points further in the final paragraphs of Discussion.

The genetic analyses are reasonable, but could be considerably strengthened. The below suggestions are some examples – it is not the case that all of the suggested additional analyses are required for publication, but some additional work to strengthen the genetic component of the work (which is, obviously, very important) will be needed.4) First, it would be useful to further replicate associations with expression using other expression datasets (e.g., common mind, depression genes and network study) and note whether any are brain region (i.e., tissue type) specific. Further, testing whether genomic risk for these brain aging phenotypes is correlated with gene expression using a technique such as TWAS or predixcan would further add informativeness and novelty to the GWAS-aspect of this paper (see: https://bogdan.dgsom.ucla.edu/pages/twas/ and/or: https://github.com/hakyimlab/PrediXcan). Second, additional follow-up annotation of GWAS results would greatly improve this manuscript. In particular, FUMA (https://fuma.ctglab.nl/) is an invaluable and easy to use resource. It would be useful to conduct more comprehensive analyses (e.g., gene-based testing) and annotation (chromain, more distal eQTLsof GWAS findings) of GWAS results.5) You currently note several phenotypes that identified SNPs are correlated with. Much of this from the UK biobank – while this is interesting, it would be useful to explore associations in other datasets as relying on UK biobank results could be somewhat circular. An online tool – GWAS atlas PheWAS has been developed for this purpose: https://atlas.ctglab.nl/PheWAS.6) It is now widely accepted that complex behavioral phenotypes, including neural ones (e.g., PMID: 26854805) are incredibly polygenic – consistent with such data, it would be useful to consider moving beyond probing single variant associations and more comprehensively examine whether the polygenic architecture of these brain age phenotypes is correlated with other phenotypes within the literature (e.g., longevity, Alzheimer's, obesity, etc.) using LD score regression (https://github.com/bulik/ldsc). The LD hub online tool may be useful for such analyses: http://ldsc.broadinstitute.org/. It would also be useful to report SNP-based heritability for these different phenotypes.

Many thanks for these detailed suggestions – we agree that they add interesting data to the genetics in this work. We have now added a comprehensive set of additional results (see Materials and methods; subsection “Data and code availability, and additional supplementary tables and figures”; Results):

– RSIDs for all indel variants reported as peak associations (that is, all peaks for all modes and all mode-clusters).

– List of SNPs/variants in LD with peak associations (via FUMA).

– Closest-mapped gene for all LD SNPs, along with functional consequence/annotation of the SNPs (FUMA+ANNOVAR).

– Chromatin state information for all LD SNPs.

– eQTL mapping for all LD SNPs and all genes, from GTEx and a range of other eQTL databases.

– Chromatin interactions for all genomic loci.

– PHEWAS – details of other traits from previous studies having associations with our peak SNPs.

– Genetic (SNP polygenic) (co)heritability of all modes and mode-clusters using LD score regression, and with Alzheimer’s disease and Parkinson’s disease.